# The confidence-noise confidence-boost (CNCB) model of confidence rating data

Pascal Mamassian[1]*, Vincent de Gardelle[2]

**1** Laboratoire des systèmes perceptifs, Département d'études cognitives, École normale supérieure, PSL University, CNRS, Paris, France, **2** CNRS and Paris School of Economics, Paris, France

* pascal.mamassian@ens.fr

## Abstract

Over the last decade, different approaches have been proposed to interpret confidence rating judgments obtained after perceptual decisions. One very popular approach is to compute meta-d' which is a global measure of the sensibility to discriminate the confidence rating distributions for correct and incorrect perceptual decisions. Here, we propose a generative model of confidence based on two main parameters, confidence noise and confidence boost, that we call *CNCB model*. Confidence noise impairs confidence judgements above and beyond how sensory noise affects perceptual sensitivity. The confidence boost parameter reflects whether confidence uses the same information that was used for perceptual decisions, or some new information. This CNCB model offers a principled way to estimate a confidence efficiency measure that is a theory-driven alternative to the popular M-ratio. We then describe two scenarios to estimate the confidence boost parameter, one where the experiment uses more than two confidence levels, the other where the experiment uses more than two stimulus strengths. We also extend the model to experiments using continuous confidence ratings and describe how the model can be fitted without binning these ratings. The continuous confidence model includes a non-linear mapping between objective and subjective confidence probabilities that can be estimated. Altogether, the CNCB model should help interpret confidence rating data at a deeper level. This manuscript is accompanied by a toolbox that will allow researchers to estimate all the parameters of the CNCB model in confidence ratings datasets. Some examples of re-analyses of previous datasets are provided in S1 File.

## Author summary

After each perceptual decision, humans have the ability to rate how confident they are that their decision is correct. While there is a growing number of models that attempt to explain how confidence judgments are built, there are very few objective measures of the sensitivity of these confidence judgments. We offer

**Data availability statement:** The Matlab toolbox that implements the CNCB model is available here: Mamassian P. CNCB toolbox (Version 0.1). Zenodo, 2024. doi:10.5281/zenodo.13348146 The simulations presented in the paper are available here: Mamassian P. Simulations of the CNCB model for the analysis of confidence ratings data. Zenodo, 2025. doi:10.5281/zenodo.14632514

**Funding:** This work was supported by French grants from the Agence Nationale de la Recherche (ANR-10-BLAN-1910 "Visual Confidence" to PM; ANR-18-CE28-0015 "VICONTE" to PM and VdG; ANR-17-EURE-0017 to PM). The funders had no role in study design, data collection and analysis, decision to publish, or preparation of the manuscript.

**Competing interests:** The authors have declared that no competing interests exist.

here a measure of confidence efficiency whose interpretation is simple. It takes a value of zero when participants are unable to evaluate the validity of their perceptual decision, a value of one when participants use the same information for their perceptual decision and their confidence judgment, and a value larger than one when participants use information for their confidence judgment that was not used for their perceptual decision. This measure of confidence efficiency is based on a generative model called CNCB that has two main parameters, confidence noise and confidence boost. These parameters again have a simple interpretation, and we show under which circumstances they can be estimated on their own. Finally, we extend the model to continuous confidence ratings where participants are not restricted to use a limited set of confidence levels.

## Introduction

As humans, our sensory and cognitive capacities are limited, and thus we can only construct imperfect representations of our environment. Consequently, we cannot avoid making some errors when judging external stimuli. However, even when we make a mistake, we may still have a sense of the precision of our perceptual judgment, and we can evaluate the probability that our response was correct. Such evaluations are judgments of confidence and are called Type 2 judgments, in contrast to Type 1 decisions which pertain to the primary perceptual task. Confidence judgments are a simple but essential expression of metacognition, that is the ability to evaluate and regulate our own mental processes [1]. They are most common in our daily life and they have been of interest since the early days of experimental psychology [2]. In the last decade, there has been an interest in modelling confidence performance, with a variety of approaches (for a recent comparison of these models, see [3]), but with no clear answer to the question of how to formally define and measure the efficiency with which human observers produce confidence judgments. In the present work, we address this question by specifying a generic process model for confidence ratings, in which we can define the confidence of an ideal metacognitive observer and the efficiency of human metacognitive abilities in comparison to this ideal observer [4,5].

Within the framework of Signal Detection Theory (SDT [6,7]), many studies have collected confidence ratings for the purpose of building Receiver Operating Characteristic (ROC) curves to evaluate Type 1 performance. However, these studies did not investigate how these confidence ratings were produced, they only assumed that observers used the same internal axis of evidence that they used for the Type 1 decision, and just placed additional criteria to produce graded confidence ratings. Other studies have focused on confidence itself, and its relation to decision accuracy, using metrics such as calibration (the root mean squared error between confidence and accuracy), resolution (the average confidence in correct responses minus confidence for errors), or the correlation between confidence and accuracy, which are easy to implement and to interpret but intrinsically confounded with Type

1 performance (see, e.g., [8,9]). To address this limitation, Maniscalco and Lau [10] have proposed meta-d' as a measure of the sensitivity of confidence ratings, in the same units as Type 1 sensitivity. The M-ratio, defined as the ratio of meta-d' over d', has been suggested to be a measure of efficiency of confidence ratings [11], unconfounded with Type 1 performance, unlike previous measures [12]. This approach has become popular in studies of metacognition, particularly in perception and memory research. However, it suffers from two key limitations.

First, the generative process that is assumed for confidence ratings has not been explicitly clarified in this framework. As a consequence, values of M-ratio cannot be easily mapped onto specific internal parameters and in particular values greater than 1 are hard to interpret. Second, the conditions of application of the meta-d' measure are highly restricted: meta-d' can only be estimated precisely within a single difficulty level. Indeed, applying this measure to a dataset with multiple levels of difficulty leads to an artefactual inflation of metacognitive skills (see for instance [13]), as participants may have more information at their disposal than what is assumed in the analysis.

We propose a novel framework for quantifying metacognitive efficiency from confidence ratings, which addresses both issues. Our approach involves a fully specified generative model for confidence [4], which is based on elements of Signal Detection Theory [6]. In this framework, perceptual decisions are based on the comparison of some sensory evidence to a sensory criterion. Similarly, it is convenient to model confidence judgments as based on some confidence evidence, and to characterize how this confidence evidence is derived from sensory evidence. For the ideal confidence observer, who uses exactly the same information for her perceptual decisions and confidence judgments, there is a one-to-one mapping between sensory and confidence evidence. Individuals may however deviate from this ideal confidence observer in many ways, and here we highlight in particular two computational mechanisms. First, as usually assumed, observers can be specifically impaired in their evaluation of confidence, above and beyond their limited sensitivity in the perceptual decision. This loss of information can be modelled as confidence noise. Second, observers can use information for their confidence judgment that was not used while they computed their perceptual decision. We call this additional information the confidence boost. While our model shares some features with other models, in particular the CASANDRE model [14] that can also fit simultaneously multiple stimulus strengths and that also emphasizes the criticality of a good estimate of sensory noise to compute confidence, the introduction of a confidence boost is specific to our approach. It is interesting to think how confidence boost could be manipulated experimentally, for instance by varying the delay between Type 1 and Type 2 responses [15].

As our approach allows for an identification of both mechanisms, we call this model the Confidence-Noise Confidence-Boost model (hereafter CNCB). Importantly, by specifying the generative model, we are able to model confidence judgments in empirical datasets that may involve varying levels of task difficulty, and quantify metacognitive efficiency across difficulty levels. As we demonstrate below, this framework is applicable to both discrete confidence ratings and continuous estimations on a (possibly distorted) probability scale. By doing so, our approach opens possibilities to develop model-based formulations of overconfidence, and to bridge a gap between metacognition research and the literature on judgment and decision making.

## Results

Because we intend to compare the CNCB model to the M-ratio, we first place ourselves in the conditions of the M-ratio. The M-ratio is the ratio of meta-d' over d', where d' is the actual sensory sensitivity and meta-d' is the theoretical d' that is consistent with the confidence judgments [10]. This latter analysis imposes the constraint that there are only two possible stimuli. For instance, the stimuli could be some dots moving to the right or to the left, that we represent by their respective stimulus strengths $\mu_s = +u$ and $\mu_s = -u$, and the observer's task is to indicate whether dots are moving to the right or to the left. Here the two stimulus strengths have the same magnitude $u$. We also consider for now that there are only two confidence levels (high and low). We release these two constraints later.

## A generative model for perceptual decisions and confidence ratings

Our generative model of decision and confidence can be summarized in two steps [4]. The first step determines the Type 1 behaviour, that is what the observer reports as their percept given the stimulus that is presented. We are guided here by classical Signal Detection Theory [6] whereby the observer has access to some sensory evidence $s$ that is simply the stimulus strength corrupted by sensory noise $\epsilon_s \sim N\left(0, \ \sigma_s^2\right)$

$$s = \mu_s + \epsilon_s. \tag{1}$$

The Type 1 decision $D$ is just the comparison of this sensory evidence to a sensory criterion $\theta_s$, and it is categorical (e.g., $+1$ for 'Right' or $-1$ for 'Left', see **Fig 1A**)

$$D = \text{sign}\left(s - \theta_s\right). \tag{2}$$

The second step determines the Type 2 behaviour, that is what the observer reports as their confidence that their perceptual decision is correct, separately for each stimulus and each perceptual decision (**Fig 1B**). In the same vein as for the Type 1 step, we assume that the observer has access to some confidence evidence, noted $w$. In our framework, this confidence evidence is based on the initial sensory evidence $s$, with weight $(1 - \alpha)$, and on additional information from the stimulus $\mu_s$ with weight $\alpha$. We call $\alpha$ the boost parameter, which varies between 0 and 1. A value of 0 corresponds to a direct link between sensory evidence and confidence evidence, whereas a value of 1 corresponds to a perfect access to the stimulus, bypassing the impact of sensory noise on confidence judgments (we previously referred to this extreme case as the super-ideal confidence observer [4]). Intuitively, if the experimenter were to reveal the identity of the stimulus to the participant after their perceptual decision but before their confidence judgment, then these confidence judgments would be consistent with a model where the confidence boost is 1. Confidence noise is modelled as noise added to the confidence evidence that by default we take to be Gaussian, $\epsilon_c \sim N\left(0, \ \sigma_c^2\right)$. Combining confidence noise and confidence boost produces confidence evidence

$$w = \left(\mu_s + (1 - \alpha) \cdot \epsilon_s - \theta_s\right)/\sigma_s + \epsilon_c. \tag{3}$$

In this equation, the sensory noise $\epsilon_s$ is the same as in **Eq. 1**. Importantly, sensory evidence is normalized by sensory noise $\sigma_s$ to make confidence evidence an estimate of performance in units of internal noise, and thus independent of the sensory dimension being evaluated [4]. Sensory and confidence evidence together define a bivariate normal distribution when both sensory and confidence noise samples are normally distributed (**Fig 1C**, left plot). A confidence boundary can then be drawn on the confidence evidence axis to separate high and low confidence judgments [16].

To be a proper Type 2 judgment, one also needs to take into account the actual perceptual decision in the trial. Indeed, although they should mostly align, confidence evidence may occasionally contradict the actual perceptual decision. When this is the case, the confidence evidence is evidence that runs against the observer's initial decision. To formalize this, one can model the Type 2 judgment $C$ as based on the product of the perceptual decision and the confidence evidence [4] as follows:

$$\begin{cases} C = \text{High if } D \cdot w > b, \\ C = \text{Low if } D \cdot w \leq b \end{cases} \tag{4}$$

where $b$ is the confidence boundary between high and low confidence ratings. From the experimenter's perspective, we want to evaluate the validity of the confidence judgment, that is to compare it to the correctness of the perceptual decision.

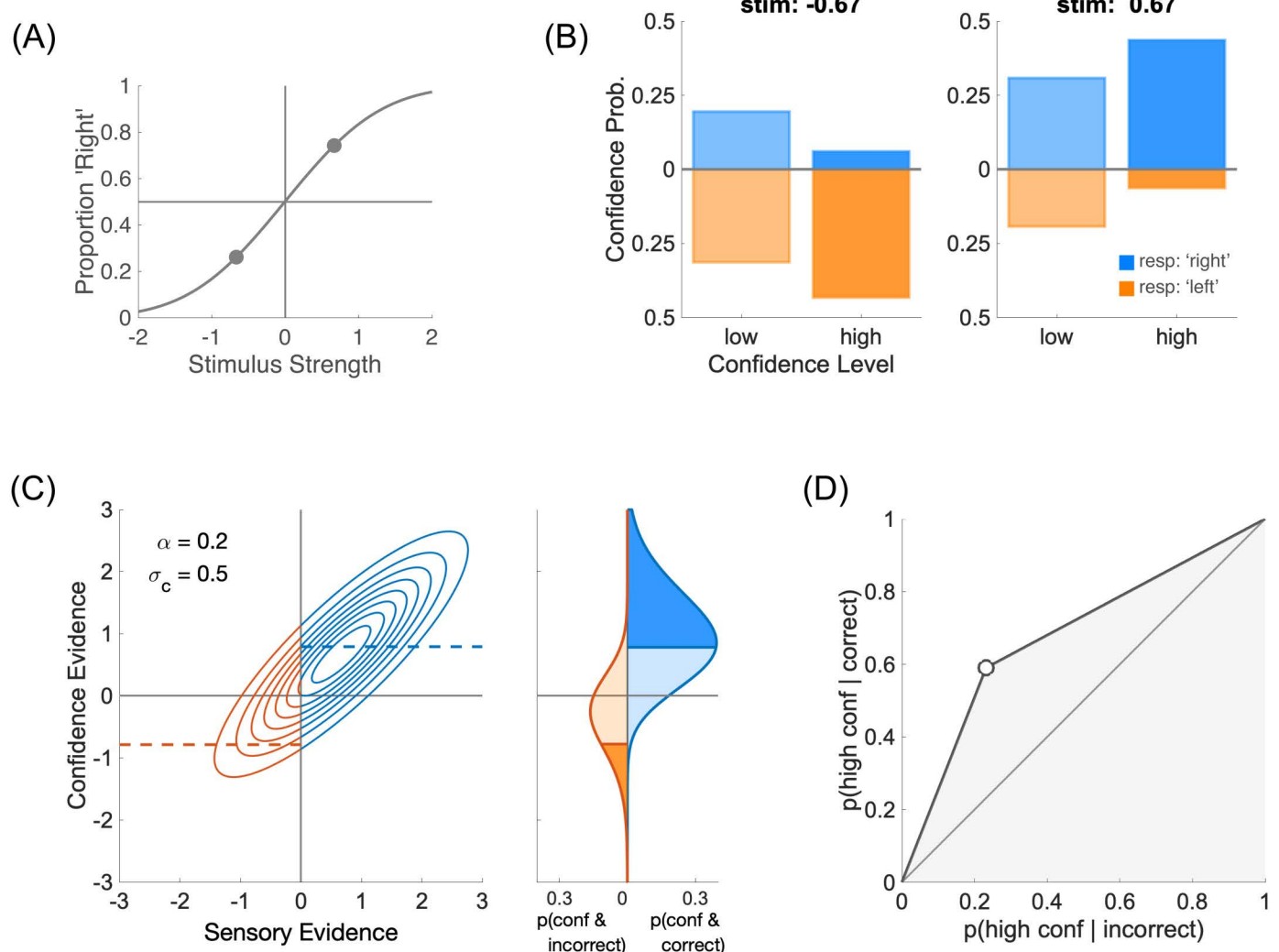

**Fig 1. Type 1 and Type 2 performance. (A)** Type 1 performance. The proportion of responding 'Right' is plotted for each of the two stimuli $\mu_s = \pm 0.67$, thus providing two points on the psychometric function. In this figure, sensory noise is $\sigma_s = 1$. **(B)** Type 2 performance. For each stimulus (left and right panels), confidence probability is plotted for each confidence level ('low' and 'high'). Blue bars represent 'Right' perceptual responses (i.e., correct when $\mu_s = +0.67$), and orange bars, running downwards, 'Left' responses (i.e., incorrect when $\mu_s = +0.67$). In this figure, confidence noise is $\sigma_c = 0.5$ and confidence boost is $\alpha = 0.2$. **(C)** Left panel: joint distribution of sensory and confidence evidence for one of the two stimuli, $\mu_s = +0.67$. When the sensory criterion is $\theta_s = 0$, perceptual responses are correct when sensory evidence is positive (blue), and incorrect when it is negative (orange). The confidence boundary separates high from low confidence judgments. It is shown as a dashed blue line for correct responses and negative dashed orange line for incorrect responses. To obtain the same fraction of high and low confidence judgments (see panels **B**), the confidence boundary was set to $b = 0.79$. Right panel: marginal confidence probability distributions for correct responses (in blue) and incorrect responses (in orange). Higher levels of confidence are shown in more saturated colours. **(D)** Type 2 ROC. The Type 2 ROC curve plots the Type 2 hit rate against the Type 2 false alarm rate. The area under the Type 2 ROC (shown in shaded grey) is a measure of confidence sensitivity.

The joint probability $P(C = \text{High} \cap \text{correct})$ of being highly confident and correct is shown on the right-hand side of **Fig 1C**. This joint probability helps us define the Type 2 hit rate $P(C = \text{High} \mid \text{correct}) = P(C = \text{High} \cap \text{correct}) / P(\text{correct})$, that is the probability of making a high confidence judgment when the perceptual decision was correct. Likewise, we can define the Type 2 false alarm rate

$$\begin{cases} \mathrm{Hit}_2(b) = \ \mathrm{P}\left(C = \mathrm{High} \mid \mathrm{correct}\right) \ = \mathrm{P}\left(Dw > b \mid D = \mathrm{sign}\left(\mu_s\right)\right) \\ \mathrm{FA}_2(b) = \ \mathrm{P}\left(C = \mathrm{High} \mid \mathrm{incorrect}\right) = \mathrm{P}\left(Dw > b \mid D \neq \mathrm{sign}\left(\mu_s\right)\right) \end{cases}.$$

(5)

The Type 2 hit and false alarm rates then define the Type 2 ROC ([17,18]; **Fig 1D**). The area under the Type 2 ROC is an index of confidence sensitivity (larger is better).

### Confidence efficiency for confidence ratings

In the CNCB model, there is a trade-off between the effects of the confidence noise and confidence boost parameters: a larger confidence noise can be compensated by a larger confidence boost to produce similar confidence sensitivities. Because of this trade-off, a given point in the Type 2 ROC space can be obtained from different pairs of confidence noise and confidence boost parameters, as shown in **Fig 2A**. In fact, all these pairs that are compatible with this Type 2 ROC point can be represented as a continuous set (see **Fig 2B**), and this set can be fully characterized by taking the most extreme pair in the set, that is the one for which confidence boost is 1 (green dot in **Fig 2B**). This extreme case corresponds to a theoretical confidence observer enjoying full access to the stimulus, but still subjected to confidence noise. In this case, we call the corresponding confidence noise $\tau_{\mathrm{human}}$ the "equivalent confidence noise" for the human observer.

Confidence efficiency indicates the extent to which human participants can perform confidence judgments as well as the ideal confidence observer. The ideal confidence observer is the theoretical observer that is using exactly the same

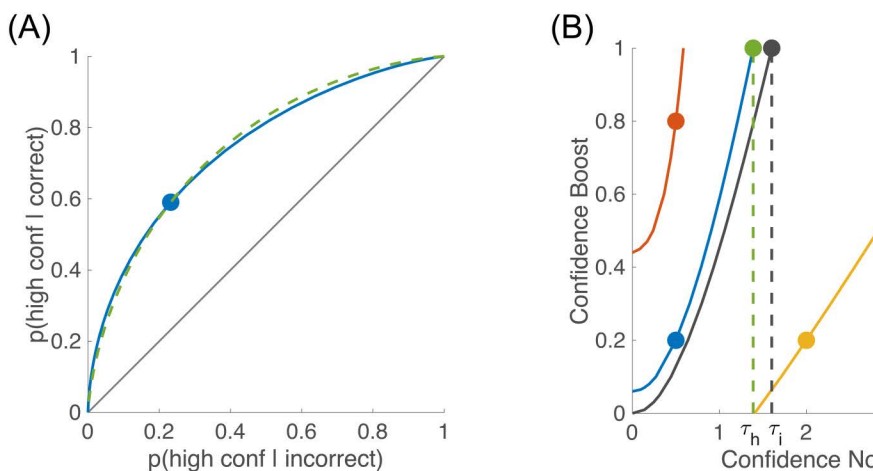

**Fig 2. Trade-off between confidence noise and confidence boost. (A)** Type 2 ROC. The blue dot shows the Type 2 hit and false alarm rates for the simulated human observer of Fig 1 ($\sigma_c = 0.5$, $\alpha = 0.2$, and $b = 0.79$). The smooth blue curve corresponds to the theoretical ROC curve obtained with the same values of confidence noise and confidence boost, but by varying the value of the confidence boundary. The dashed green curve corresponds to another observer with different confidence noise and confidence boost ($\sigma_c = 1.39$ and $\alpha = 1.0$), chosen such that Type 2 hit and false alarm rates are identical to the simulated human observer. Note that these two curves intersect in a single point (blue dot), and slightly differ outside this point. In fact, there is an infinite number of pairs of confidence noise and confidence boost that will give the same set of Type 2 hit and false alarm rates, as illustrated in panel B. **(B)** Equivalent pairs of confidence noise and confidence boost. Each coloured line represents an observer with a particular set of Type 2 hit and false alarm rates. The blue curve shows the pairs of confidence noise and confidence boost that are equivalent to the simulated human observer shown by the blue dot that was also the example shown in panel A. The black curve shows the equivalent pairs of confidence noise and confidence boost for the ideal confidence observer (i.e., for whom $\sigma_c = 0$ and $\alpha = 0$). The equivalent confidence noise is obtained when the confidence boost $\alpha$ is set to 1, and is shown as a green dot for the human observer ($\tau_h = 1.39$), and as a black dot for the ideal observer ($\tau_i = 1.60$). The red and yellow curves are equivalent pairs of confidence noise and confidence boost for other simulated observers that are presented later in this paper, namely ($\sigma_c = 0.5$, $\alpha = 0.8$) in red and ($\sigma_c = 2.0$, $\alpha = 0.2$) in yellow.

information for the perceptual decision and the confidence judgment. By definition, the ideal observer has no confidence noise ($\sigma_c = 0$) and no confidence boost ($\alpha = 0$). To best match the ideal confidence observer to the human observer, we choose confidence boundaries for the ideal that best approximate the overall human distribution of confidence judgments for the different levels. Just like the human observer, many pairs of confidence noise and confidence boost can produce the same Type 2 hit and false alarm rates (black curve in **Fig 2B**), and by taking the extreme pair with a confidence boost of 1, we can define an equivalent confidence noise for the ideal observer $\tau_{\text{ideal}}$ (black dot in **Fig 2B**).

The confidence efficiency $\eta$ is then defined as the ratio of the squared equivalent confidence noises

$$\eta = \tau_{\text{ideal}}^2 / \tau_{\text{human}}^2. \tag{6}$$

Because this measure of efficiency is obtained from the CNCB model, we refer to it as the *CNCB efficiency*. In a later section below, we compare this CNCB efficiency measure to the M-ratio.

## Disentangling confidence noise and boost with multiple confidence levels

We now describe two cases where confidence noise and confidence boost can be estimated separately. The first of these cases is when the experiment contains multiple confidence levels.

When observers have the opportunity to use $n$ levels for their confidence rating, with ($n > 2$), they have to choose ($n - 1$) confidence boundaries and compare their confidence evidence to these boundaries. As an illustration, we take the same perceptual task with two stimulus strengths (Fig 3A). For each of these stimulus strengths, we simulate an observer who is asked to rate their confidence on a 4-point scale, while trying to have the same fraction of confidence judgments for each confidence level (Fig 3B). These confidence judgments are obtained by setting three confidence boundaries along the confidence evidence internal space (Fig 3C). Therefore, when we plot Type 2 hit against false alarm rates, we obtain a Type 2 ROC curve that is now defined by three points (Fig 3D). These three points now constrain better the location of the Type 2 ROC, and thus should allow us to disentangle the confidence noise and confidence boost parameters. Indeed, whereas a single point in the Type 2 ROC space was compatible with many Type 2 ROC curves intersecting on this single point (and thus with many pairs of confidence noise and boost), specifying more than one point sets stronger constraints by which a single curve, and thus a single pair of confidence noise and boost parameters, can be identified.

We ran simulations where we vary the number of confidence levels from 2 to 6, while keeping the constraint that each confidence level contains the same fraction of confidence judgments. Importantly, we see that confidence efficiency is still well estimated even for small numbers of confidence levels (Fig 4A). To quantify the reliability of the efficiency measure, we computed the discriminability between two conditions (here the conditions where confidence noise and boost are $(0.5, \ 0.2), (0.5, \ 0.8)$). The discriminability measure is the area under the ROC to separate the distributions of efficiencies obtained from the simulations. As expected, these simulations confirm that confidence noise (Fig 4B) and confidence boost (Fig 4C) are better estimated when there are more confidence levels. In particular, confidence boost cannot be estimated reliably when there are only two confidence levels (the area under the ROC is below a 75% threshold shown as a dashed line in the upper panel). Unexpectedly, the estimation of confidence noise and boost is not completely random when there are only two confidence levels. We believe that this residual ability comes from small sensory biases inherent of each simulation that create two distinct Type 2 ROC, and that in turn constrain the confidence noise and confidence boost parameters.

The ability to identify the two confidence parameters might also depend in practice not only on the number of confidence boundaries, but also on the number of trials in each confidence bin. To address this issue, we will look at the effects of the total number of simulations and of confidence biases in later sections.

## Disentangling confidence noise and boost with multiple stimulus strengths

Another way to estimate confidence noise and confidence boost is to use multiple stimulus strengths. As an example, let us consider the scenario where there are six stimulus strengths (Fig 5A). For each stimulus strength, we can extract high and low confidence judgments (Fig 5B), separately for 'Right' (in blue) and 'Left' perceptual decisions (in orange; confidence probability runs downwards). The confidence distributions are symmetric for positive and negative stimulus strengths because we assumed that the observer adopted the optimal sensory criterion (here $\theta_s = 0.0$; Fig 5C). Because of this symmetry, the six stimulus strengths produce only three Type 2 ROC curves (Fig 5D). Even though there is a single

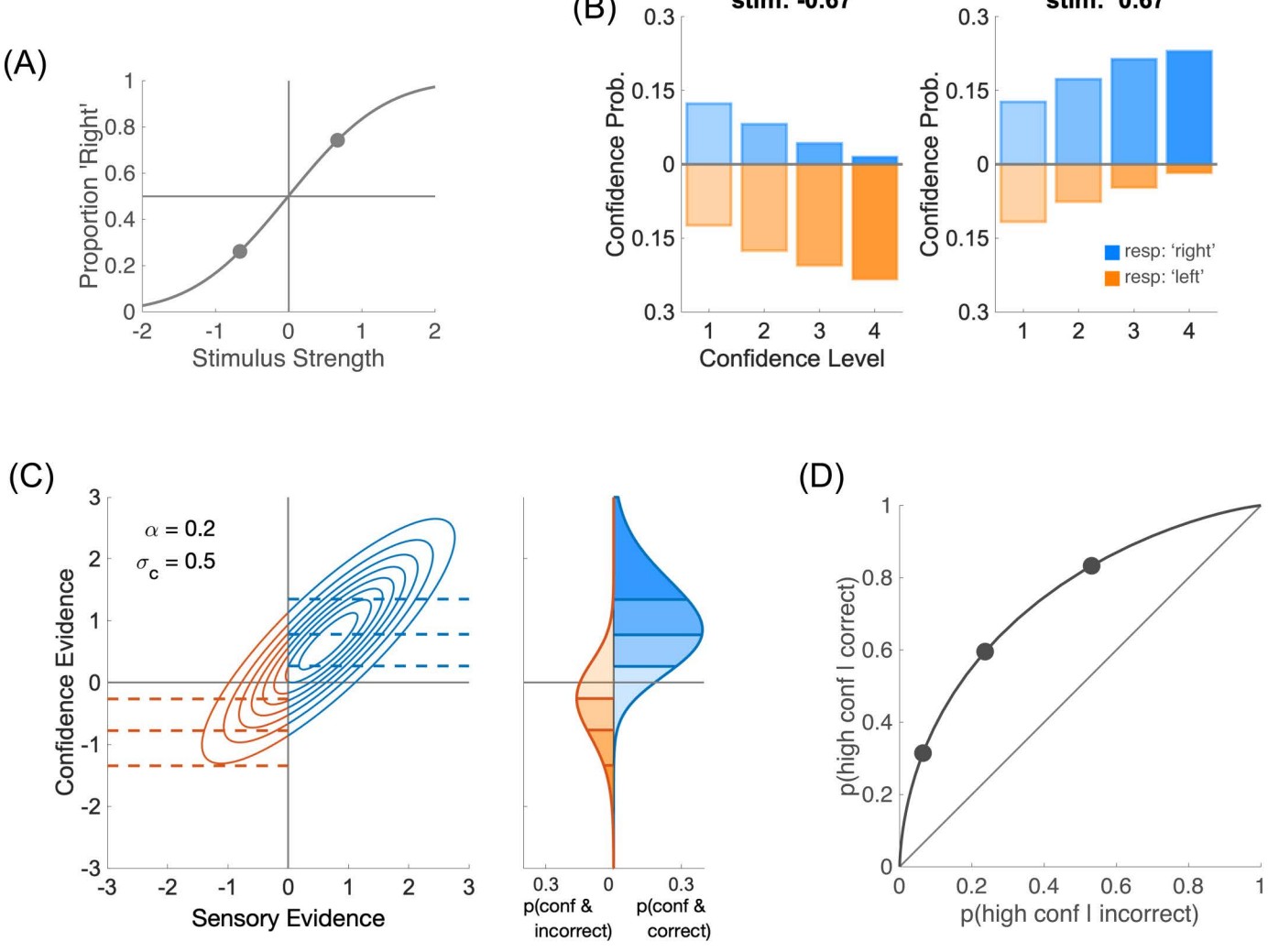

**Fig 3. Effect of multiple confidence levels on Type 2 ROC. (A)** Psychometric function from two stimulus strengths $\mu_s = \pm 0.67$, with sensory noise $\sigma_s = 1$ and no sensory bias ($\theta_s = 0.0$). **(B)** For each stimulus strength (left and right panels), confidence probability is plotted for each confidence level (1 to 4). Blue bars represent 'Right' perceptual responses, and orange bars 'Left' responses (running downwards). In this figure, confidence noise is $\sigma_c = 0.5$ and confidence boost is $\alpha = 0.2$. Confidence boundaries were chosen to produce the same fraction of confidence responses in each of the four confidence levels ($b_1 = 0.27$; $b_2 = 0.79$; $b_3 = 1.35$). **(C)** Left panel: joint distribution of sensory and confidence evidence for one of the stimulus strengths ($\mu_s = +0.67$). Perceptual responses are correct to the right of the criterion (in blue), and incorrect to its left (in orange). When confidence is judged on a 4-point scale, there are three confidence boundaries as shown by the dashed lines. Right panel: confidence probabilities for correct and incorrect responses in blue and orange, respectively. Higher levels of confidence are shown in more saturated colours. **(D)** Type 2 ROC. When confidence is rated on 4 levels, the Type 2 ROC is constrained by 3 points.

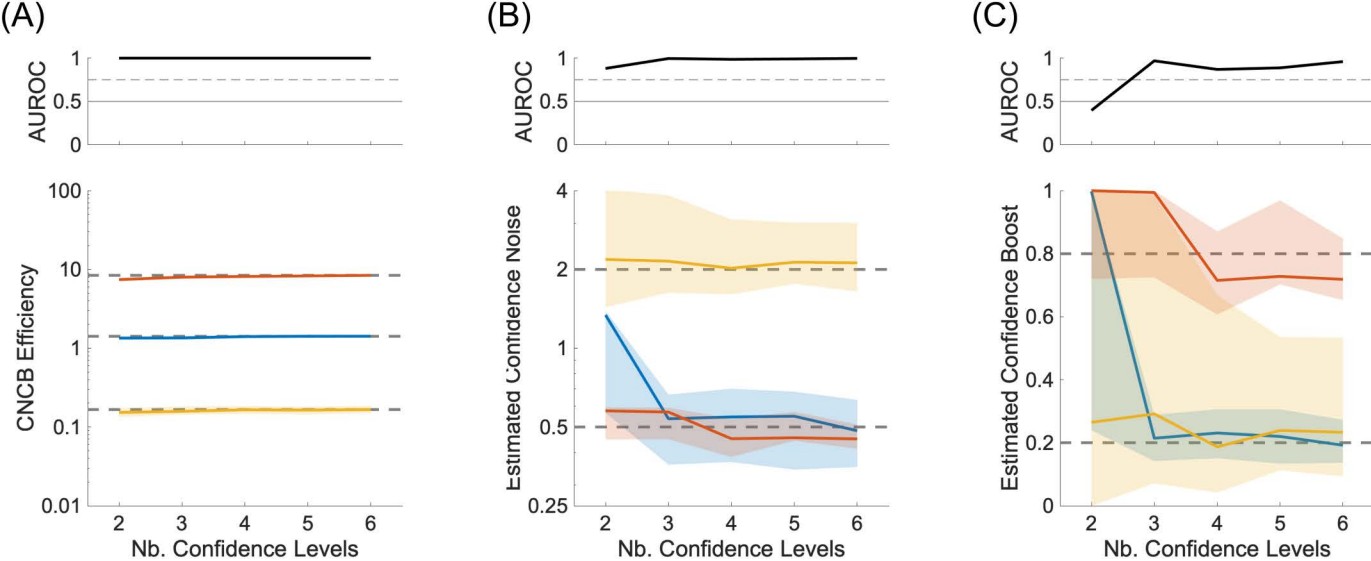

**Fig 4. Confidence noise and confidence boost from multiple confidence levels. (A)** Lower panel: CNCB efficiency estimates as a function of number of confidence levels in the simulation. The colours represent three conditions for the confidence noise and confidence boost parameters $(\sigma_c, \alpha)$, namely $(0.5, 0.2)$ in blue, $(0.5, 0.8)$ in red, and $(2.0, 0.2)$ in yellow. Upper panel: Discriminability of the blue and red simulations. In all lower panels, solid curves show median values over 100 repeated simulations, and shaded regions represent the interquartile range. The dashed lines correspond to the median CNCB efficiency obtained for the largest number of confidence levels (presumably the best estimates). Other parameters are listed in the Methods section. **(B)** Estimated confidence noise. Upper panel: Discriminability of the blue and yellow simulations. **(C)** Estimated confidence boost. Upper panel: Discriminability of the blue and red simulations.

point on each of these ROC curves (corresponding to the division between high and low confidence judgments), these three points constrain the model better than if there was a single one, and thus allow us to disentangle the confidence noise and confidence boost parameters.

To confirm this, we ran simulations where we vary the number of stimulus strengths from two to seven. We restricted the number of confidence levels to 2 ('high' and 'low') to verify that with only two stimulus strengths, confidence noise and confidence boost cannot be estimated (see Fig 2). We use a confidence criterion such that the two confidence levels are used equally within each simulation. The stimulus strengths are chosen to be equally spaced between $-2$ and $2$, with these two extreme values omitted. An example with two stimulus strengths is shown in Fig 1A, and an example with six stimulus strengths is shown in Fig 5A. Importantly, we see that confidence efficiency is not biased by the number of stimulus strengths, and that it is still well estimated even for small numbers of stimulus strengths (Fig 6A). The estimation of confidence noise (Fig 6B) and confidence boost (Fig 6C) improves with the number of stimulus strengths, as expected. In particular, confidence boost cannot be estimated when there are only two stimulus strengths. Unexpectedly, it is also not completely random when there are only two stimulus strengths, which as discussed in the previous section, may reflect small sensory biases inherent of each simulation.

We note that including several stimulus strengths is common practice in psychophysics as this allows the experimenter to measure the full psychometric function, and thus a better estimate of sensory noise. However, in the confidence literature, the common meta-d' measure of confidence sensitivity [10] can only accommodate one stimulus for each response category, and disregarding fluctuations in stimulus strengths can lead to biased estimates with this measure [13]. Our approach, by contrast, naturally accommodates multiple difficulty levels by estimating a single sensory noise, and thus takes advantage of such designs to provide more refined estimates of subcomponents of the metacognitive process. We can expect that releasing the constraint on the number of stimulus strengths will enrich the design of future experiments that use confidence ratings.

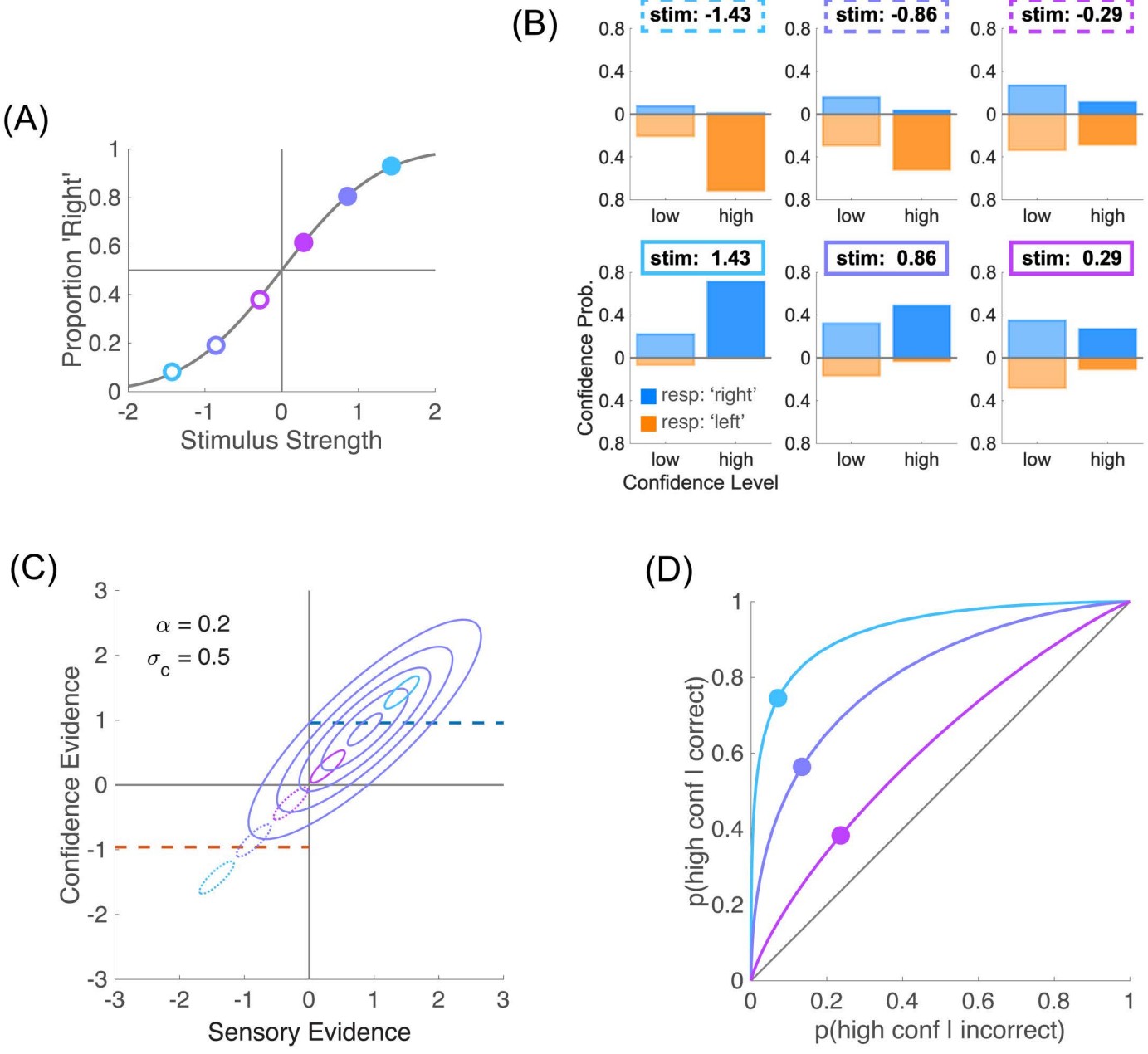

**Fig 5. Effect of multiple difficulty levels on Type 2 ROC. (A)** Psychometric function from six stimulus strengths chosen to be equally spaced between $\mu_s = -1.43$ and $\mu_s = +1.43$. In this figure, sensory noise is $\sigma_s = 1$ and sensory criterion is $\theta_s = 0.0$. **(B)** For each stimulus strength, confidence probability is plotted for each confidence level ('high' and 'low'). Blue bars represent 'Right' perceptual responses, and orange bars 'Left' responses (running downwards). In this figure, confidence noise is $\sigma_c = 0.5$ and confidence boost is $\alpha = 0.2$. **(C)** The sensory evidence for each stimulus strength has an associated confidence evidence. The full bivariate distribution of sensory and confidence evidence is shown for one of the six stimulus strengths. Positive stimulus strengths are shown in solid lines, negative ones in dotted lines. **(D)** When the observer is unbiased ($\theta_s = 0.0$), there are now three Type II ROC curves. The three dots correspond to the Type 2 hit and false alarms rates obtained when the confidence boundary was set to equate the overall fraction of high and low confidence ratings ($b = 0.96$) over all six stimulus strengths.

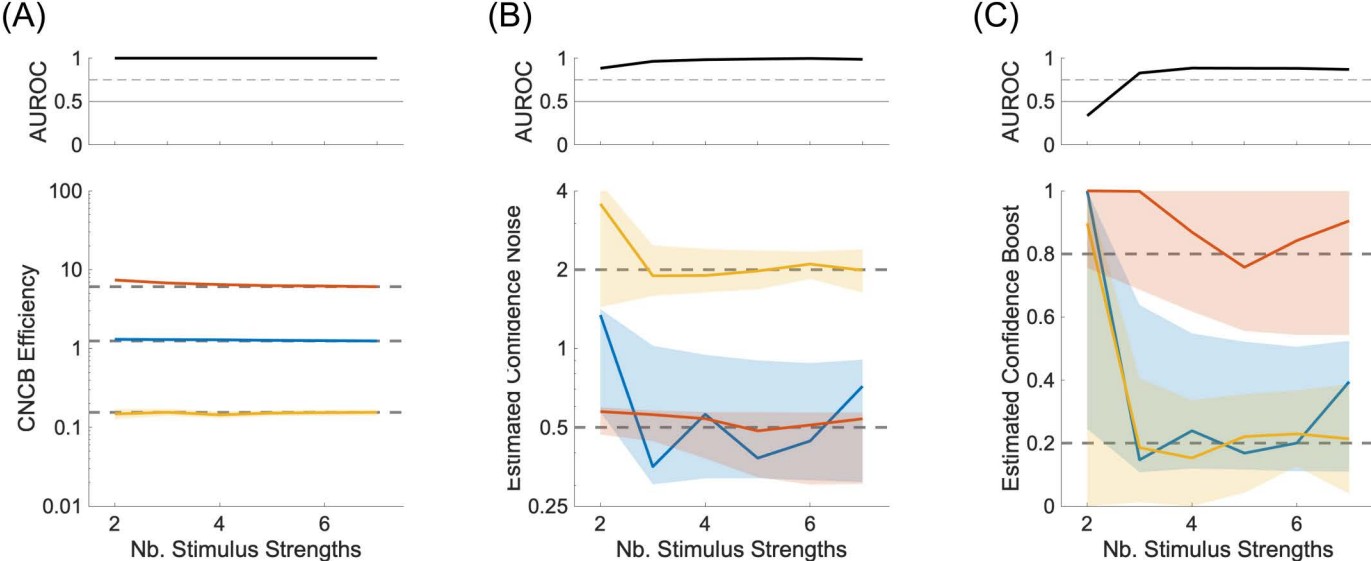

**Fig 6. Confidence noise and confidence boost from multiple stimulus strengths. (A)** Lower panel: CNCB efficiency estimates as a function of number of stimulus strengths in the simulation. The colours represent three conditions for the confidence noise and confidence boost parameters ($\sigma_c$, $\alpha$), namely $(0.5, 0.2)$ in blue, $(0.5, 0.8)$ in red, and $(2.0, 0.2)$ in yellow. Upper panel: Discriminability of the blue and red simulations. In all lower panels, solid curves show median values over 100 repeated simulations, and shaded regions represent the interquartile range. The dashed lines correspond to the median CNCB efficiency obtained for the largest number of stimulus strengths (presumably the best estimates). Other parameters are listed in the Methods section. **(B)** Estimated confidence noise. Upper panel: Discriminability of the blue and yellow simulations. **(C)** Estimated confidence boost. Upper panel: Discriminability of the blue and red simulations.

## Model recovery

The previous two sections have shown that the confidence noise and confidence boost parameters could be disentangled when the experiment involves more than two confidence levels or more than two stimulus strengths. This suggests that models with or without the confidence boost parameter could be distinguished. We now address this question more directly by performing a model comparison analysis.

We have simulated data from two types of models. Model 1 does not contain a confidence boost parameter (it is equivalent to setting this parameter to zero), and Model 2 does contain a confidence boost parameter that is drawn randomly on its domain (from 0 to 1) across simulations. In both cases, we used six stimulus strengths and 4 confidence levels. We also randomized sensory noise, sensory criterion, confidence noise, and confidence bias (see section Materials and Methods for details). We then fitted these simulated datasets by the two types of models (without and with a confidence boost parameter) and check which model best fit the data. We used likelihood ratio tests for nested models [19] to test whether Model 2 is better than Model 1. We computed the test statistic $\lambda_{\mathrm{LR}} = -2\,(\lambda_1 - \lambda_2)$, where $\lambda_1$ (resp. $\lambda_2$) is the log likelihood of Model 1 (resp. Model 2). If Model 1 is correct, then this test statistic is asymptotically distributed as a $\chi_1^2$ random variable (the degrees of freedom is 1 because the two models differ by only one parameter).

The simulations indicate that the two models are reasonably well recovered over a reasonable range of other parameters' values (Fig 7A). When Model 1 was simulated, it was recovered 99% of the times, and when Model 2 was simulated, it was recovered 76% of the times, thus resulting in a discrimination sensitivity of the two models of $d' = 3.24$, with a bias in favour of Model 1 (discrimination criterion: $c = 0.92$). The bias is expected because when Model 2 is simulated with a small confidence boost, this model is not distinguishable from Model 1 (Fig 7C, right panel). When Model 2 is simulated with a large confidence noise, there is also greater confusion with Model 1 (Fig 7C, left panel). When Model 1 is simulated, recovery of that model is very good, and estimated confidence noise and boost are also very good (Fig 7B).

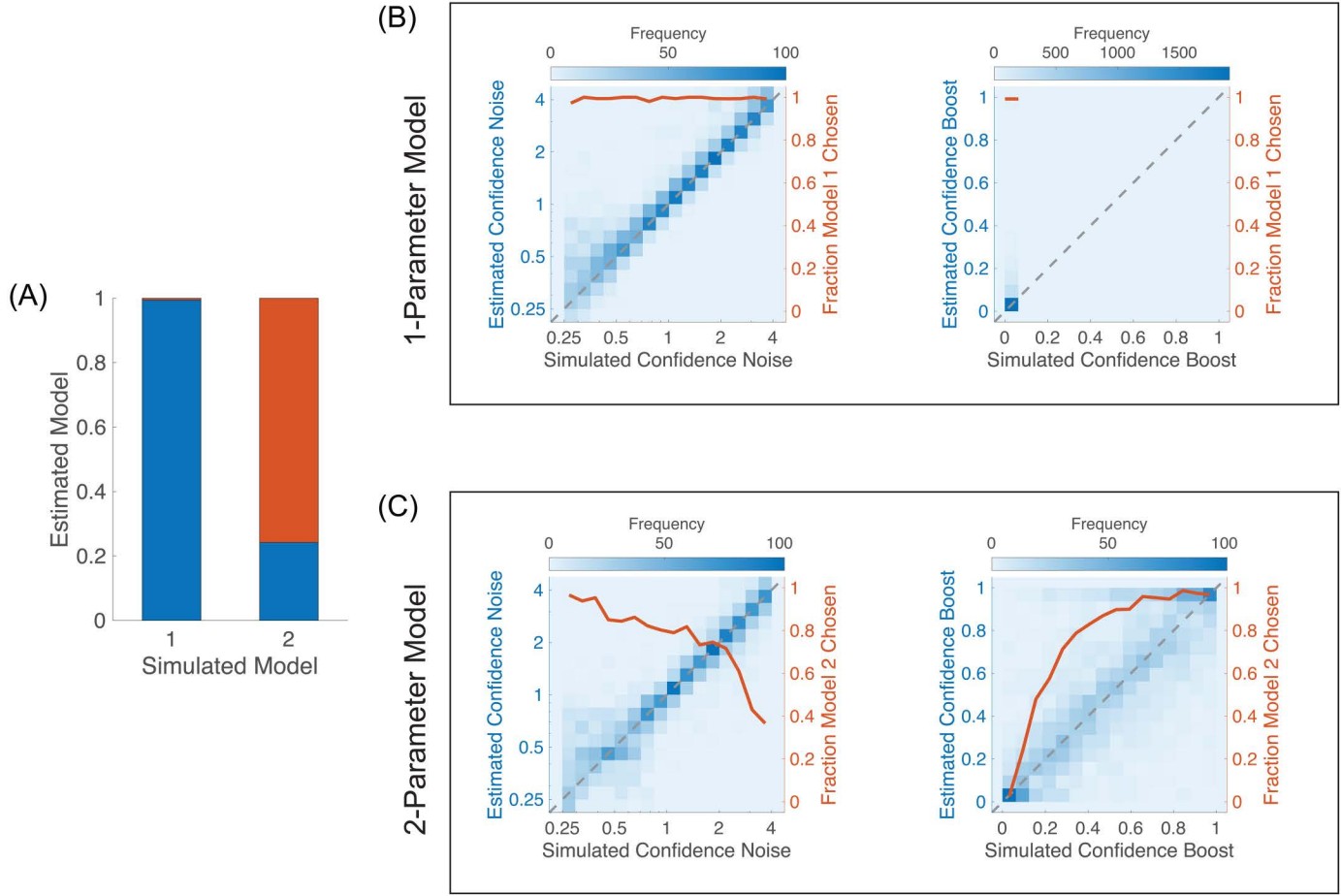

**Fig 7. Model recovery. (A)** Fraction of estimated models for each simulated model. Model 1 does not have a confidence boost parameter and Model 2 does. Blue bars correspond to fractions of estimated Model 1, and red bars Model 2. **(B)** Simulations of Model 1. Heatmaps are parameters' estimates from Model 2, with saturation showing the frequency of estimated parameter over 2,500 simulations. Left panel, left axis: estimated confidence noise as a function of simulated confidence noise. The diagonal dashed line shows perfect performance. The axis on the right and the red curve show the fraction of times Model 1 was favoured over Model 2 as a function of confidence noise. Right panel: same analysis as that in the left panel, but for confidence boost instead of confidence noise. **(C)** Simulations of Model 2. Left and right panels show the effects of simulated confidence noise and confidence boost, respectively, on the estimated confidence noise and confidence boost, and on the fraction of times Model 2 was favoured over Model 1.

From the simulations of Model 2, we note that the estimated confidence boost tends to cling to the extremes of its domain (i.e., either 0 or 1; Fig 7C, right panel). While this is a limitation of our model, we believe that in practice, experimenters will be mostly interested in testing whether confidence boost deviates or not from 0 (the condition where confidence is using the same information as the perceptual decision). In the next section, we examine more closely how well confidence noise and confidence boost can be estimated.

## Recovery of confidence noise and confidence boost

We have seen in previous sections that confidence noise and confidence boost of the CNCB model can in theory be estimated when there are more than two confidence levels, and that the CNCB efficiency can be estimated even for designs containing more than two stimulus strengths. Here, we illustrate more precisely how confidence noise and confidence boost affect CNCB efficiency and how estimated efficiency depends on the number of trials in the data, an issue that is of high practical interest.

As expected, as confidence noise increases, the CNCB efficiency decreases (Fig 8A), and both the confidence noise and confidence boost can be well recovered (Fig 8B-C). However, we observe a small bias in the recovery of the confidence parameters when both confidence noise and confidence boost are very small, suggesting that the confidence noise parameter sometimes absorbs more of the variability in the confidence judgments than necessary. Also as expected, as confidence boost increases, the CNCB efficiency increases (Fig 8D), and again, both confidence noise and confidence boost can be well recovered (Fig 8E-F).

Increasing the number of confidence judgments naturally increases the precision of the estimated confidence efficiency and the CNCB parameters (Fig 8G-I). From these simulations, we can observe that a good estimation of confidence efficiency is already available with 100 confidence judgments (Fig 8G). These simulations also suggest that a good estimation of confidence noise require at least 200 confidence judgments, and confidence boost parameter about 2,000 confidence judgments (Fig 8H-I). These quick observations would benefit from a more thorough analysis that goes beyond the scope of the present paper.

### Influence of sensory parameters

In this section, we investigate how the Type 1 sensitivity and bias affects the recovery of the Type 2 parameters. To manipulate Type 1 sensitivity, we varied the sensory noise over a range that led to a probability of correct perceptual decisions over $[0.55, 0.95]$ if the sensory criterion was optimal. We can see that both the M-ratio and our CNCB efficiency measures are less precise and less accurate when the probability of being correct is above 0.85 or below 0.6 (Fig 9A-B). Likewise, the recovery of the confidence noise and boost parameters are better when the probability of being correct is below 0.90 or above 0.6 (Fig 9C-D).

Perceptual biases arise from sensory criteria that are not placed at their optimal location (Fig 10A). It is known that M-ratios are not fully independent of sensory criteria (see Fig 6C in [12]; Fig 10B). This is due in part to the fact that when there is a bias in sensory criterion, each sensory strength generates its own Type 2 ROC curve, and the meta-d' computation needs to find a way to integrate these two incompatible sets of confidence into a single model. In fact, since the original work of Maniscalco & Lau [10], there have been other proposals to compute meta-d' (see for instance the definition of $\widetilde{d'}_b$ in [12]). In contrast, the CNCB model naturally takes into account the sensory criterion because there is only one set of sensory parameters that tries to explain all the perceptual responses and all the confidence judgments (see Eq. 3). Under the same simulated conditions, CNCB confidence efficiency appears to be more robust to biases in sensory criteria (Fig 10C). We note also that confidence noise (Fig 10D) and confidence boost (Fig 10E) can be recovered well irrespective of the sensory criteria.

### Influence of confidence bias

So far, we have assumed that the confidence judgments were distributed uniformly across all confidence levels. This helped minimize the possibility that some confidence levels were never visited for some combinations of stimulus strengths and perceptual decisions. We relax here this constraint to look at the effect of confidence biases on the recovery of the confidence parameters. To simulate the effect of a confidence bias, we replaced the uniform distribution of confidence levels by a triangular distribution, where the number of confidence judgments increases (or decreases) linearly across the confidence levels. The parameter $\delta$ describes the ratio of confidence judgments between the top and bottom levels. If it is larger than $1$, there will be a dominance of high confidence values. The simulations shown in Fig 11 indicate that the recovery of confidence noise and boost is quite robust over a large range of confidence biases.

### Comparison of CNCB efficiency and M-ratio

Since both the CNCB efficiency and the M-ratio measure derive from elements of Signal Detection Theory, although following slightly different routes, it is natural to think that they should be related. In this part, to compare CNCB efficiency

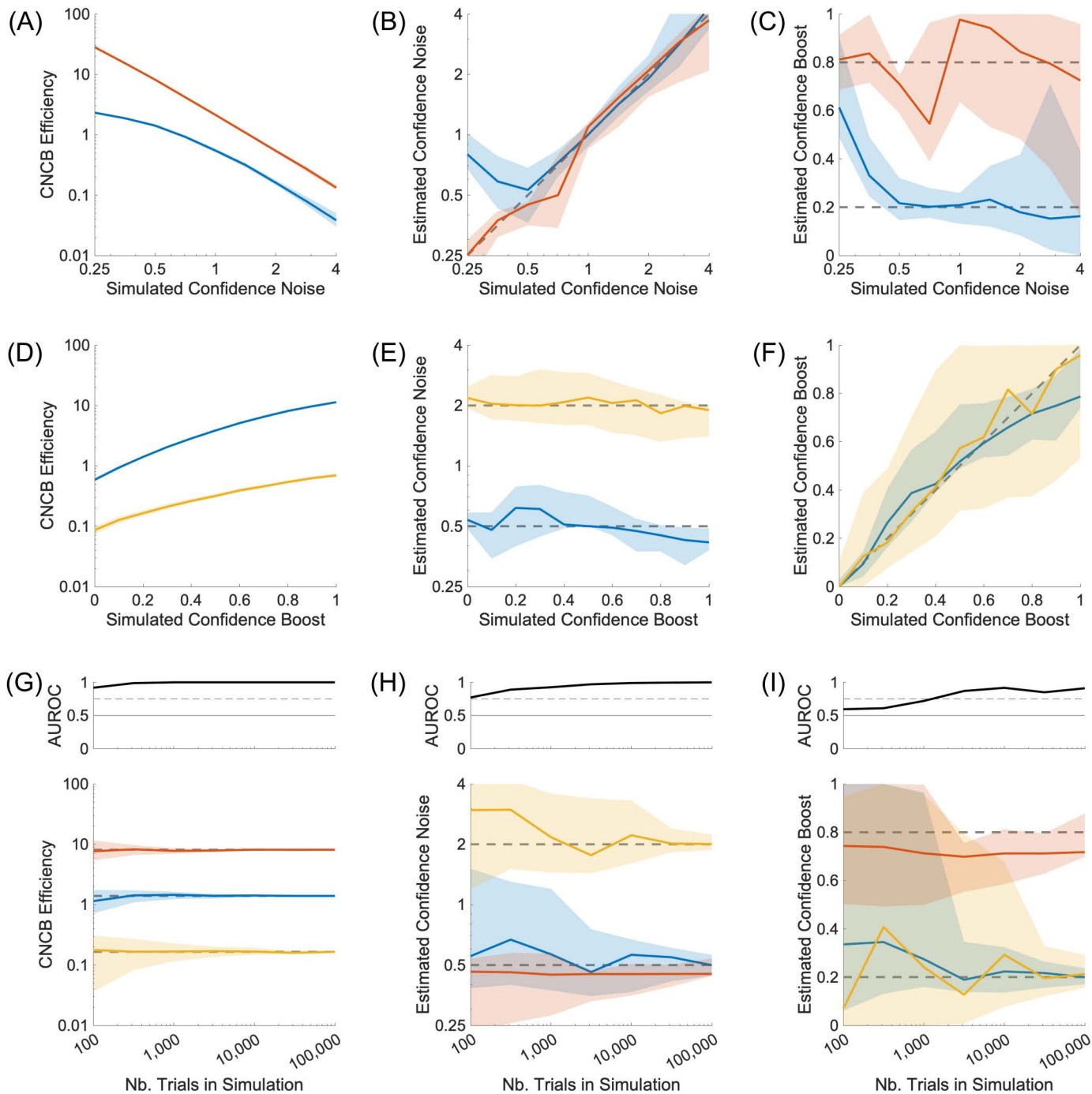

**Fig 8. Parameters recovery for varying confidence noise, confidence boost, and number of confidence judgments. (A-C)** Effect of different values of simulated confidence noise, for two different values of confidence boost ($\alpha = 0.2$ in blue, and $\alpha = 0.8$ in red). **(D-F)** Effect of different values of simulated confidence boost, for two different values of confidence noise ($\sigma_c = 0.5$ in blue, and $\sigma_c = 2.0$ in yellow). **(G-I)** Effect of the number of confidence judgments in the simulation, for three pairs of confidence noise and confidence boost parameters ($\sigma_c$, $\alpha$), namely $(0.5,\ 0.2)$ in blue, $(0.5,\ 0.8)$ in red, and $(2.0,\ 0.2)$ in yellow. Upper panels show discriminability of the blue and red simulations (G), blue and yellow (H), and blue and red (I). Panels show estimated confidence efficiency (A, D, G), estimated confidence noise (B, E, H), and estimated confidence boost (C, F, I). In lower panel (G), the dashed lines correspond to the median CNCB efficiency obtained for the largest number of confidence judgments (presumably the best estimates). In all panels, solid curves show median values over 100 repeated simulations, and shaded regions represent the interquartile range. Unless specified otherwise, simulations consisted in 10,000 confidence judgments. Other parameters are listed in the Methods section.

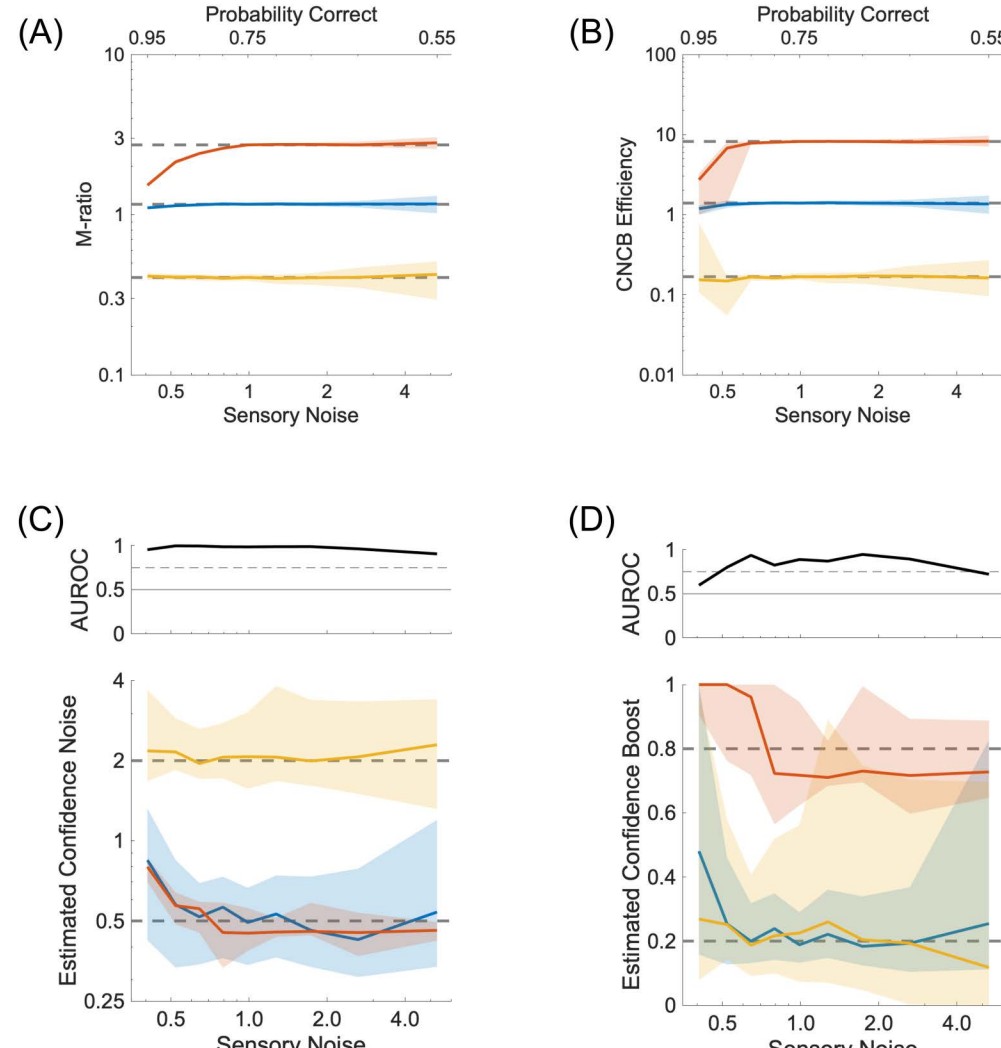

**Fig 9. Effect of sensory sensitivity. (A)** M-ratio as a function of different values of sensory noise. The colours represent three conditions for the confidence noise and confidence boost parameters $(\sigma_c, \alpha)$, namely $(0.5, 0.2)$ in blue, $(0.5, 0.8)$ in red, and $(2.0, 0.2)$ in yellow. Continuous lines and shaded areas show the medians and interquartile ranges, respectively, over 100 repeated simulations. The dashed lines correspond to the median M-ratio obtained for the sensory noise that has been used in the other simulations $(\sigma_s = 1.0)$. **(B)** CNCB efficiency estimates for the same conditions as in (A). **(C)** Estimated confidence noise. Upper panel: Discriminability of the blue and yellow simulations. **(D)** Estimated confidence boost. Upper panel: Discriminability of the blue and red simulations. Other parameters are listed in the Methods section.

and M-ratio, we limit ourselves to only 2 stimulus strengths ($\mu_s = \pm 1$). From simulations where we varied confidence noise, confidence boost, and the number of confidence judgments (Fig 8), we find that there is a one-to-one relationship between CNCB efficiency and M-ratio (Fig 12)

$$\eta \approx (\text{M-ratio})^2 . \qquad (7)$$

The square in the relationship is expected given that d' is inversely related to the standard deviation of the sensory noise (and similarly for meta-d') whereas the CNCB efficiency is a ratio of squared equivalent noises. This relationship makes sense when one varies confidence noise because confidence noise has similar effects on the meta-d' and the CNCB

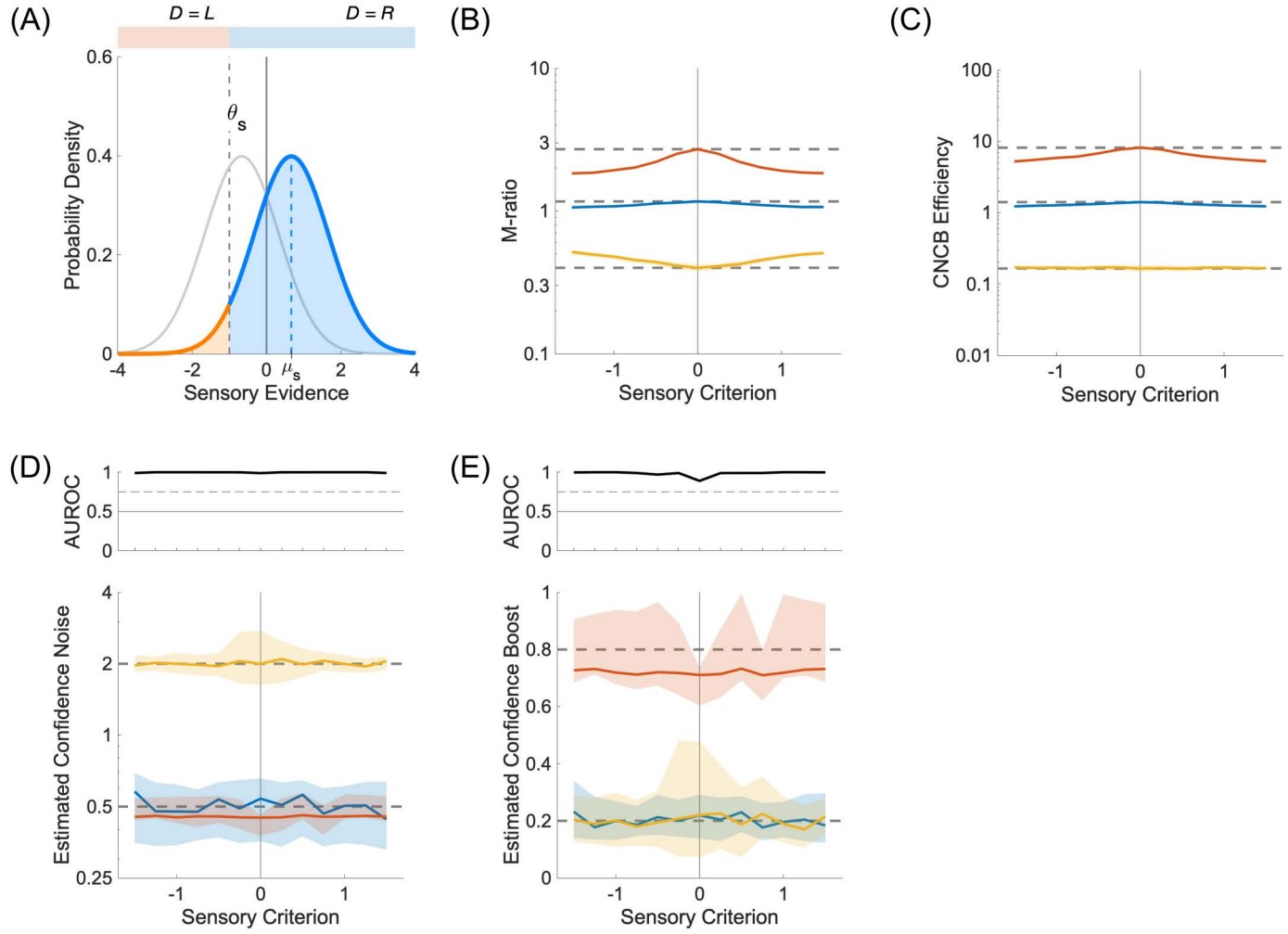

**Fig 10. Effect of sensory criteria. (A)** Probability densities for sensory strengths $\mu_s = \pm 0.67$. Perceptual decisions are made relative to a biased sensory criterion (here, $\theta_s = -1.0$). **(B)** M-ratio as a function of different values of sensory criteria. The colours represent three conditions for the confidence noise and confidence boost parameters $(\sigma_c, \alpha)$, namely $(0.5, 0.2)$ in blue, $(0.5, 0.8)$ in red, and $(2.0, 0.2)$ in yellow. Continuous lines and shaded areas show the medians and interquartile ranges, respectively, over 100 repeated simulations. The dashed lines correspond to the median M-ratio obtained for the optimal sensory criterion ($\theta_s = 0.0$), presumably providing the best estimates. The M-ratio is strongly biased by non-optimal sensory criteria. **(C)** CNCB efficiency estimates for the same conditions as in (B). The CNCB efficiency seems stable across sensory criteria. **(D)** Lower panel: Estimated confidence noise. Upper panel: Discriminability of the blue and yellow simulations. **(E)** Lower panel: Estimated confidence boost. Upper panel: Discriminability of the blue and red simulations. Other parameters are listed in the Methods section.

model. Interestingly, the relationship seems to hold very well over a large range of model parameters (Fig 12). Further theoretical work is necessary to find a closed form derivation for the equivalence between CNCB efficiency and M-ratio.

## Continuous confidence ratings

As the number of confidence levels increases, one approaches the extreme case where confidence is rated on a continuous scale. This scenario is typically obtained when observers are asked to use a continuous scale, say between 0 and 100, to report their subjective probability that their perceptual decision was correct. This special case deserves its own treatment because one needs to model how subjective probabilities are possibly biased.

When estimating probabilities, human participants are often over-estimating small probabilities and under-estimating large probabilities. To model this kind of biases, we adhere to the proposal of Zhang & Maloney [20] to use the following family of probability distortions. In Eqs 8 and 9 and in the text, the function

$$Lo\left(\pi(p)\right) = \gamma\, Lo(p) + (1 - \gamma)\, Lo\left(p_0\right) \tag{8}$$

where $p$ denotes the true (objective) probability, $\pi(p)$ denotes the corresponding distorted (subjective) probability, $p_0$ is a constant (the fixed point of the transformation), and $Lo$ is the log-odds function

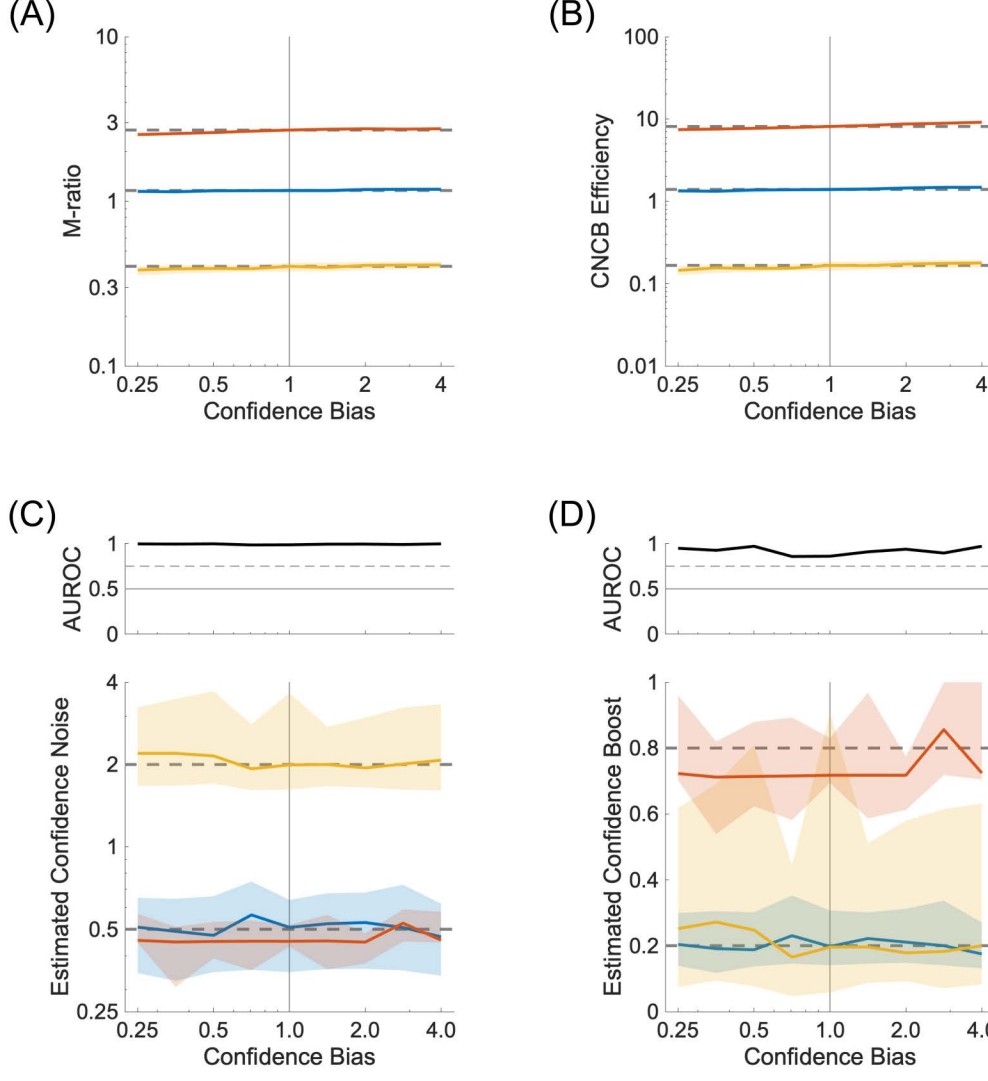

**Fig 11. Effect of confidence bias. (A)** M-ratio as a function of different values of confidence bias. The colours represent three conditions for the confidence noise and confidence boost parameters $(\sigma_c,\ \alpha)$, namely $(0.5,\ 0.2)$ in blue, $(0.5,\ 0.8)$ in red, and $(2.0,\ 0.2)$ in yellow. Continuous lines and shaded areas show the medians and interquartile ranges, respectively, over 100 repeated simulations. The dashed lines correspond to the median M-ratio obtained for a uniform distribution of confidence judgments across all levels, presumably leading to the least bias in M-ratio estimate. **(B)** CNCB efficiency estimates for the same conditions as in (A). **(C)** Estimated confidence noise. Upper panel: Discriminability of the blue and yellow simulations. **(D)** Estimated confidence boost. Upper panel: Discriminability of the blue and red simulations. Other parameters are listed in the Methods section.

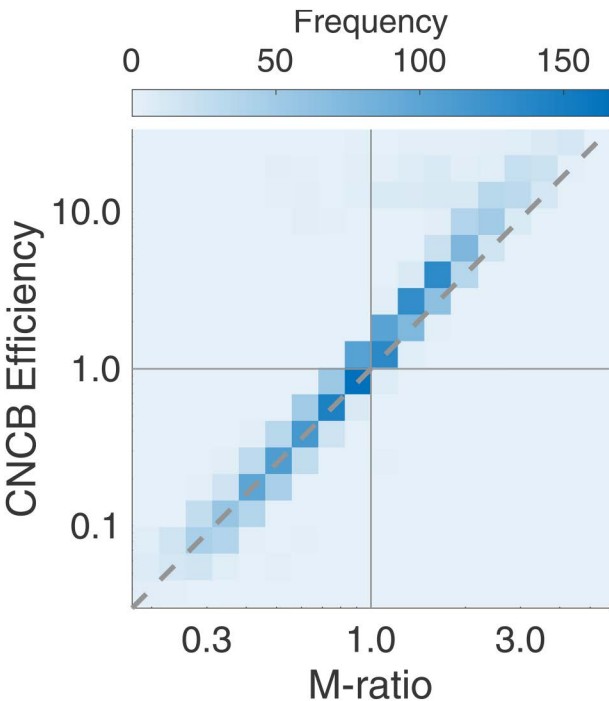

**Fig 12. Relationship between confidence efficiency and M-ratio. (A)** The heatmap shows CNCB efficiencies and M-ratios from 2,500 simulated experiments where there were only two stimulus strengths and four confidence levels. Other parameters were sampled from the ranges described in the Materials and Methods section. The diagonal shows a squared relationship between M-ratio and confidence efficiency in log-log coordinates (see Eq. 7).

$$Lo(p) = \log\left(\frac{p}{1-p}\right). \tag{9}$$

The parameter $\gamma$ in this family of distortions allows us to capture an overestimation of small probabilities and underestimation of large probabilities (when $\gamma < 1$) or the reverse pattern (when $\gamma > 1$). Default parameter values that produce a non-biased relationship between objective and subjective probabilities are $\gamma = 1$ and $p_0 = 0.5$ (Fig 13A). This non-biased relationship produces highly skewed distributions of confidence ratings towards high confidence (Fig 13B).

Using this family of probability distortions is a compact way to model the infinite number of confidence boundaries using only two parameters ($\gamma$ and $p_0$). An example of such probability distortions is shown in Fig 13C, and the associated distribution of confidence ratings is shown in Fig 13D.

When such probability distortions are introduced between objective and subjective confidence judgments, the CNCB efficiency can be estimated nonetheless, and it is robust to variations in the distortion parameter (Fig 14A). Importantly, confidence noise and confidence boost can also be recovered very well, independently of $\gamma$ (Fig 14B and 14C), both when small probabilities were over-estimated ($\gamma < 1$) and large probabilities were over-estimated ($\gamma > 1$). The very good parameter recovery comes from the fact that the confidence evaluation has finer resolution than if confidence was restricted to a small number of confidence levels. However, in practice, it is important to check that when instructed to report their confidence on a continuous scale, participants make an effort to report their confidence judgments precisely, rather than simply using two or three confidence levels ("very sure", "not sure at all", and possibly "something in-between"). Finally, the parameters of the probability distortion ($\gamma$ and $p_0$) can be recovered very well in all the simulations (Fig 14D and 14E).

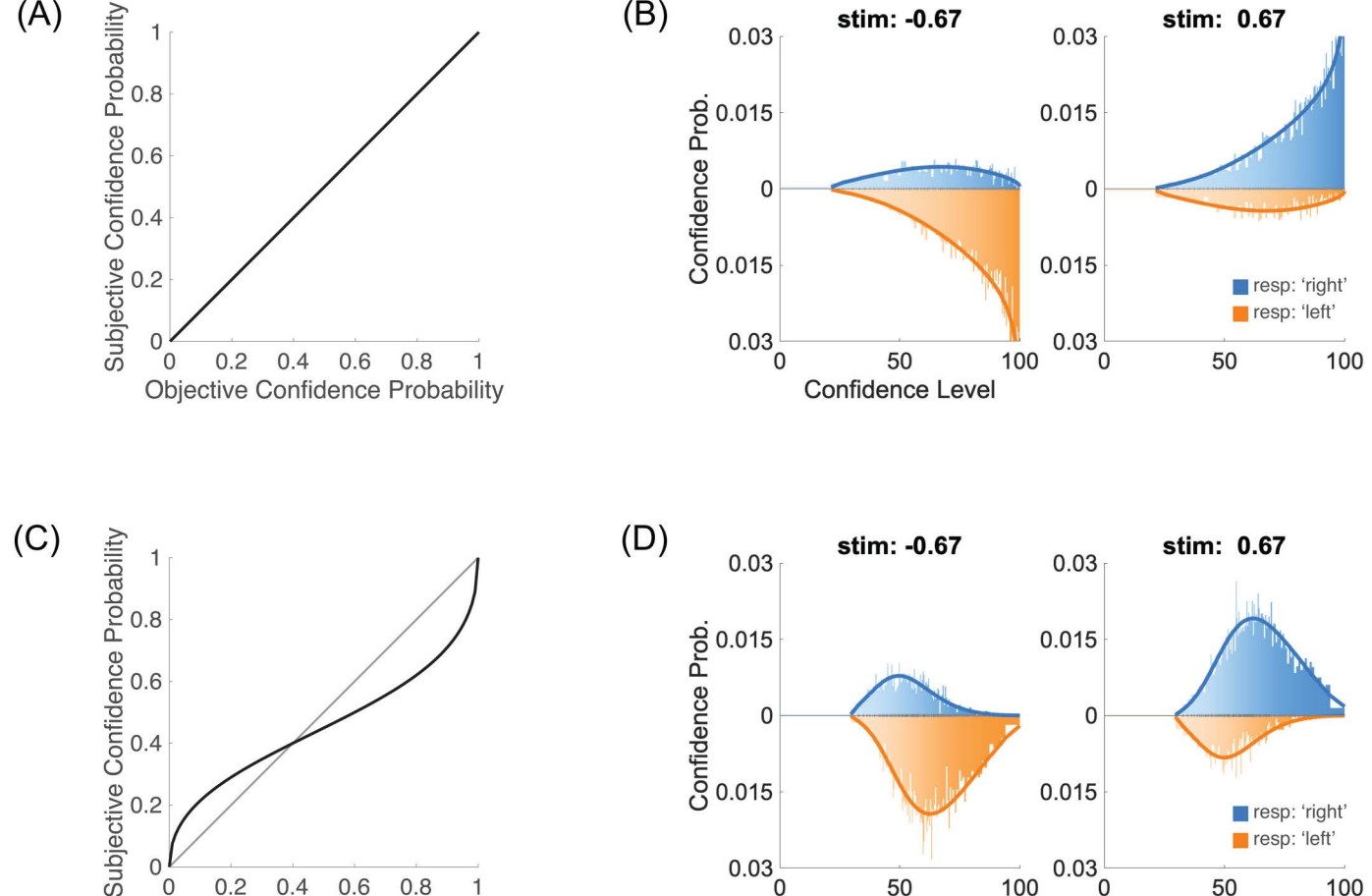

**Fig 13. Continuous confidence ratings. (A)** Non-biased relationship between objective and subjective confidence probabilities. **(B)** Confidence ratings for the two categories of stimuli ($\mu_s = \pm0.67$). The blue distribution corresponds to perceptual responses 'Right' (i.e., correct when $\mu_s = +0.67$, and incorrect when $\mu_s = -0.67$), and the orange distributions to responses 'Left' (running downwards). With the non-biased relationship in (A), confidence ratings are strongly skewed towards high confidence levels. For these simulations, $\sigma_c = 0.5$ and $\alpha = 0.2$. **(C)** Log-odds transformation between objective and subjective confidence probabilities with $\gamma = 0.5$ and $p_0 = 0.4$. **(D)** With the biased relationship in (C), and the same confidence noise and confidence boost parameters as in (B), confidence ratings are more concentrated in the middle of the range of confidence levels. Other parameters are listed in the Methods section.

## General discussion

In a world full of uncertainty, being able to evaluate the quality of one's own decisions is an essential skill to optimize behaviour. For instance, observers use their sense of confidence to guide how much effort they will put in a task [21–23], or which task they will prioritize [24,25]. Evaluations of confidence are also useful when combining information from multiple sources: to optimize a collective decision, individuals should weigh their private information by the confidence they have in it [26,27].

Yet, evaluating oneself is also challenging. Confidence judgments can be biased towards underconfidence or overconfidence [28], and individuals greatly vary in their ability to predict the accuracy of their decisions (see, e.g., [29]). Importantly, whereas the issue of overconfidence has triggered a large interest, well beyond cognitive psychology (e.g., [30]), the notion of metacognitive sensitivity has been so far less influential.

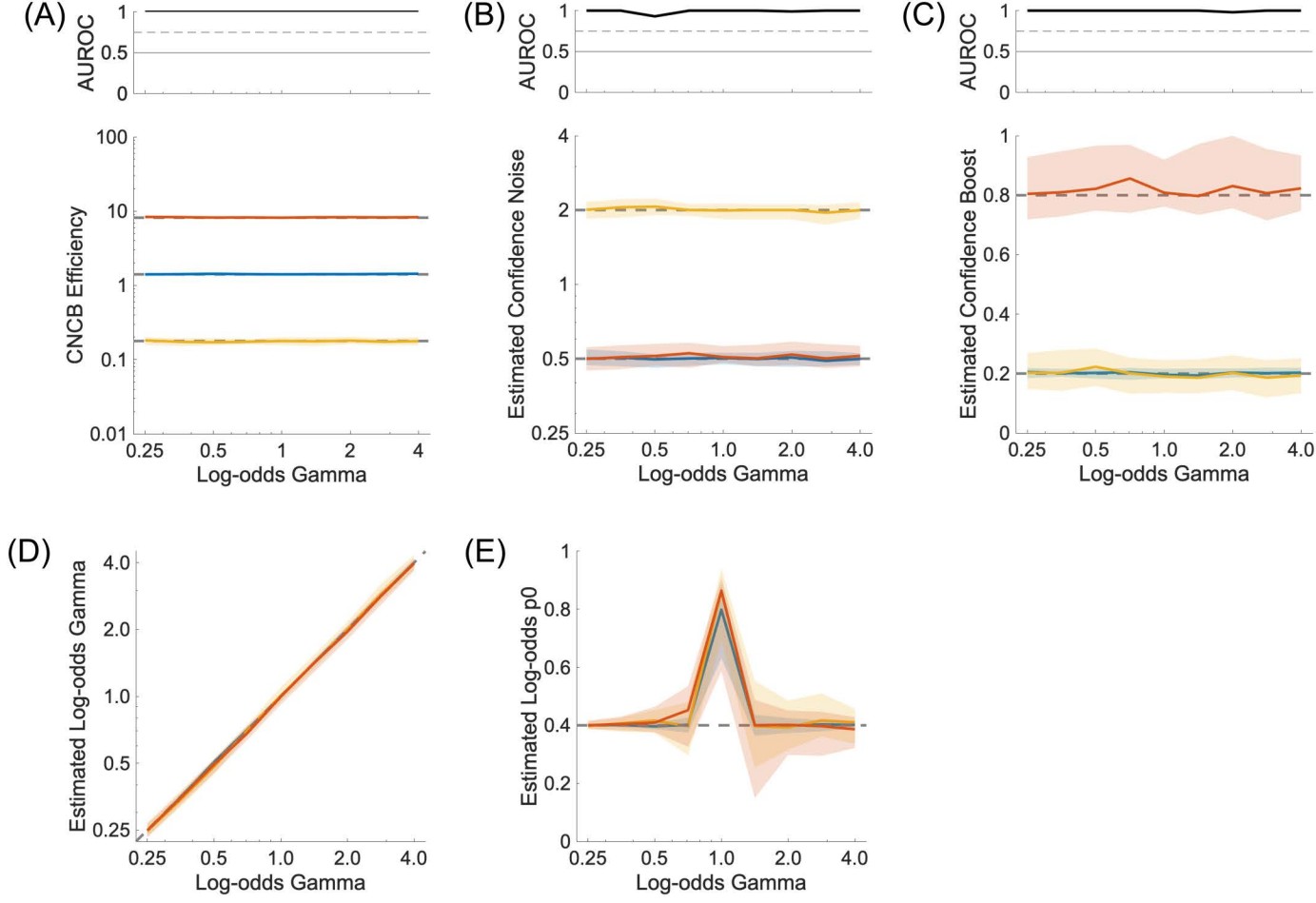

**Fig 14. Recovery and influence of probability transformation. (A)** Lower panel: Confidence efficiency for different values of the $\gamma$ parameter. Upper panel: Discriminability of the blue and red simulations. **(B)** Estimated confidence noise. Upper panel: Discriminability of the blue and yellow simulations. **(C)** Estimated confidence boost. Upper panel: Discriminability of the blue and red simulations. **(D)** Recovery of the $\gamma$ parameter of the log-odds transformation. **(E)** Recovery of the $p_0$ parameter of the log-odds transformation for different values of the $\gamma$ parameter. When $\gamma = 1$, the parameter $p_0$ is undetermined. The colours represent three conditions for the confidence noise and confidence boost parameters ($\sigma_c$, $\alpha$), namely $(0.5, 0.2)$ in blue, $(0.5, 0.8)$ in red, and $(2.0, 0.2)$ in yellow. In all panels, solid curves show median values over 100 repeated simulations, and shaded regions represent the interquartile range. Other parameters are listed in the Methods section.

One possible reason may have been the difficulty to converge on a single measure of confidence sensitivity in the prior literature, in comparison to overconfidence bias which can be simply calculated as the average confidence minus the average performance for each individual. The introduction of the meta-d' measure [10], by addressing the confound between metacognitive sensitivity and performance, presented a significant improvement on this matter. However, constraints remained regarding the designs in which this measure can be applied. We believe that allowing researchers to release the constraints of a single magnitude of difficulty level and of a finite confidence scale when analysing metacognitive sensitivity is an important contribution of the present work. In addition, our approach does not require that separate models be fitted to each response class, and this may be the reason why the CNCB model is more robust to perceptual biases than meta-d'.

By relying on a fully specified generative model of confidence judgments, our method also allows for a principled definition of the efficiency with which observers make confidence ratings. The CNCB model was introduced in our prior work to account for a particular paradigm to study confidence, the confidence forced choice [4]. Here, we extend its application to

model confidence ratings on discrete or continuous scales, allowing the analysis of various experimental paradigms within a common framework. This effort to develop a common modelling framework across experimental paradigms is considered an important objective in the domain of metacognition [31]. One possible direction for future work to extend further the generality of the present approach could be to expand the CNCB model to accommodate for more than 2 choice options at the Type 1 level, and possibly to continuous Type 1 responses. General Recognition Theory already exists as a generalization of Signal Detection Theory to multiple response categories [32], and considering situations with more than two choices has been instrumental in discriminating between theories of confidence [33]. Another possible direction is to refine the present modeling approach, by considering non-Gaussian confidence noise [34–36], or confidence noise that would reflect a mis-estimation of the sensory noise that we used to normalize our confidence evidence in Eq. 3.

In the present work, we have emphasized confidence noise and confidence boost as important parameters in our model, which both impact metacognitive efficiency but in opposite directions. We have described the conditions under which the resulting parameter indeterminacy can be lifted, such that one can separately estimate confidence noise and confidence boost. These involve having more than two confidence levels or more than one magnitude of stimulus strength. In practice, using three confidence levels or four stimulus strengths (i.e., two magnitudes for each response class) seem sufficient. The confidence boost parameter is more difficult to evaluate than the confidence noise, and good estimates seem to require at least 2,000 confidence judgments. Separating confidence noise from confidence boost should allow researchers to characterize variations of metacognition more precisely. For instance, prior research has shown that metacognitive efficiency can be degraded under stress [37] or under complex multitasking [38,39]. Conversely, it can be improved after meditation training [40], or simply by asking observers to re-evaluate their initial confidence judgment [41]. One could wonder whether such phenomena are mediated by a change in confidence noise only, in confidence boost only, or whether they involve changes in both parameters. It will be also interesting to reanalyze previous datasets with the current framework (see some examples in S1 File).

More generally, specifying the generative model for confidence judgments may allow researchers to ask novel questions or to shed new light on past topics about the computational bases of confidence judgments. For instance, one recurring question has been whether confidence is calculated before or after the decision is made, and many studies have provided evidence for post-decisional contributions to confidence (e.g., [15,42–45]). Early measures of confidence efficiency, in particular the M-ratio, offered only a global measure of confidence performance that merged together different sources of inefficiencies [46]. Recent efforts have started to disentangle these different sources (e.g., [47]). The CNCB approach presented here is in this perspective, offering a tool to separate two very different components that affect confidence efficiency (reflected by the confidence noise and the confidence boost parameters), while at the same time linking these factors to a global measure (the CNCB confidence efficiency).

To conclude, we have presented a computational model that allows to measure the efficiency of confidence ratings while varying the difficulty of the perceptual task. Our approach is set up in the domain of perceptual decisions. We hope that it can inspire other domains (see [48]) in which confidence judgments are of interest.

## Materials and methods

### Parameter values for the figures

Unless otherwise noted, all the simulations in this manuscript were obtained with the following parameters. The two sensory levels were $\mu_s = \pm 0.67$, sensory noise was $\sigma_s = 1.0$, and the sensory criterion was placed at the optimal location $\theta_s = 0.0$. These values were chosen so that they lead to 75% correct, a targeted level of performance that is often chosen by experimenters. Confidence noise was set by default to $\sigma_c = 0.5$ and confidence boost to $\alpha = 0.2$. The number of confidence levels was set to 4, with confidence boundaries chosen to equate the number of confidence choices in each confidence level. A simulated experiment was based on 10,000 trials, and plots show median and inter-quartile ranges over 100 such simulated experiments. To reduce the likelihood of local minima, the fits started with 4 different initial values for the confidence noise and confidence boost, obtained by crossing two values of $\sigma_c = \{0.5,\ 2.0\}$ and two values of $\alpha = \{0.25,\ 0.75\}$, and the best fit was chosen to be the one that had the maximum log-likelihood.

In Figs 7 and 12, we ran simulations over a range of parameters. Sensory noise $\sigma_s$ varied over a range that led to a probability of correct perceptual decisions over $[0.55,\ 0.95]$ if the sensory criterion was optimal, the sensory criterion $\theta_s$ varied over $[-1.5,\ 1.5]$, confidence noise $\sigma_c$ over $[0.25,\ 4.0]$ uniformly in log space, confidence boost $\alpha$ over $[0,\ 1]$, and confidence bias $\delta$ over $[0.25,\ 4.0]$ uniformly in log space. In Fig 7, stimulus strengths were $\mu_s = \{-1.43,\ -0.86,\ -0,29,\ 0.29,\ 0.86,\ 1.43\}$ and there were 4 confidence levels, whereas in Fig 12, stimulus strengths were $\mu_s = \{-0.67,\ 0.67\}$ and there were 4 confidence levels.

## Fitting procedure

With real data that are necessarily noisy, there will never be a perfect match between the predictions of the model and the data. For each stimulus strength $i$, and each perceptual decision $j$ on this stimulus, we note $n_{i,j,\ k}$ the number of human confidence judgments rated to be equal to $k$, where $k$ varies between $1$ and $m$. We are fitting the conditional probabilities $P(j,\ k \mid i)$ of making a perceptual decision $j$ and a confidence judgment $k$ whenever stimulus strength $i$ is presented. If we note $N_i = \sum_j \sum_{k=1}^{m} n_{i,j,\ k}$, In we have

$$P(j,\ k \mid i) = \frac{n_{i,j,\ k}}{N_i}.$$

(10)

We are trying to best match the distributions of confidence ratings $P(j,\ k \mid i)$ between the human data and the model. We find the best match by maximum likelihood estimation. The likelihood that the model produces a probability $P(j,\ k \mid i)$ is given in Eq. 11

$$L_i(j,\ k) = n_{i,j,\ k}\ log\left(P(j,\ k \mid i)\right).$$

(11)

We then try to maximize the total likelihood $L = \sum_i \sum_j \sum_{k=1}^{m} L_i(j,\ k)$. A useful metrics for the goodness of fit is the Deviance obtained from computing the likelihood of the saturated model (i.e., the upper bound on the likelihood derived from using the actual human data as a model). In Eq. 12 the second,

$$L_s = \sum_i \sum_j \sum_{k=1}^{m} \left[\, n_{i,j,\ k}\ \ \left(\log\left(n_{i,j,\ k}\right) - log\left(N_i\right)\right)\,\right].$$

(12)

The Deviance is then $G^2 = -2\,(L - L_s)$, with lower values indicating better fits. The Deviance is approximately distributed as a $\chi^2$ statistics with degrees of freedom equal to the difference in the number of parameters between the saturated and the CNCB models. A good model is one that is not statistically different from the saturated model.

## Full generative model

In our previous work [4], we have considered two additional parameters, namely $\beta$ a confidence bias parameter (the underestimation of sensory noise in the scaling of confidence evidence), and $\theta_c$ a confidence criterion (the bias for confidence evidence relative to the sensory criterion). We have disregarded these two parameters here for simplicity, but they might be relevant in some cases (for a thorough discussion of these parameters, see [4]).

## Self-consistency

In the main text, Type 2 hit and false alarm rates are defined relative to the correctness of the perceptual decision. However, correctness is arguably not the best goal that participants are trying to estimate, since without feedback, participants have no way to know that they achieved this goal. Instead of correctness, we might prefer to use self-consistency that refers to the stability of a perceptual decision [4,49]. In that case, Eq. 5 above becomes

$$\begin{cases} \text{Hit}_2(b) = \text{ P}\left(C = \text{high} \mid \text{self-consistent}\right) = \text{P}\left(D.w > b \mid D = \text{sign}\left(\mu_s - \theta_s\right)\right) \\ \text{FA}_2(b) = \text{ P}\left(C = \text{high} \mid \text{self-inconsistent}\right) = \text{P}\left(D.w > b \mid D \neq \text{sign}\left(\mu_s - \theta_s\right)\right) \end{cases}. \tag{13}$$

## Supporting information

**S1 File. Reanalysis of published datasets with the CNCB model.**
(PDF)

## Acknowledgments

The Matlab toolbox that implements the CNCB model is available here [50]: https://github.com/mamassian/cncb/. The code and data files for the simulations in this manuscript are open access [51].

## Author contributions

**Conceptualization:** Pascal Mamassian, Vincent de Gardelle.

**Formal analysis:** Pascal Mamassian, Vincent de Gardelle.

**Funding acquisition:** Pascal Mamassian.

**Investigation:** Pascal Mamassian.

**Methodology:** Pascal Mamassian.

**Project administration:** Pascal Mamassian.

**Resources:** Pascal Mamassian.

**Software:** Pascal Mamassian.

**Visualization:** Pascal Mamassian.

**Writing – original draft:** Pascal Mamassian, Vincent de Gardelle.

**Writing – review & editing:** Pascal Mamassian, Vincent de Gardelle.

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
