## [Decision Letter · Decision Letter 0]

7 Nov 2024

PCOMPBIOL-D-24-01485The Confidence-Noise Confidence-Boost (CNCB) model of confidence rating dataPLOS Computational Biology Dear Dr. Mamassian, Thank you for submitting your manuscript to PLOS Computational Biology. After careful consideration, we feel that it has merit but does not fully meet PLOS Computational Biology's publication criteria as it currently stands. Therefore, we invite you to submit a revised version of the manuscript that addresses the points raised during the review process. Please submit your revised manuscript within 60 days Jan 07 2025 11:59PM. If you will need more time than this to complete your revisions, please reply to this message or contact the journal office at ploscompbiol@plos.org. Please include the following items when submitting your revised manuscript: * A rebuttal letter that responds to each point raised by the editor and reviewer(s). You should upload this letter as a separate file labeled 'Response to Reviewers'. This file does not need to include responses to formatting updates and technical items listed in the 'Journal Requirements' section below.* A marked-up copy of your manuscript that highlights changes made to the original version. You should upload this as a separate file labeled 'Revised Manuscript with Track Changes'.* An unmarked version of your revised paper without tracked changes. You should upload this as a separate file labeled 'Manuscript'. If you would like to make changes to your financial disclosure, competing interests statement, or data availability statement, please make these updates within the submission form at the time of resubmission. Guidelines for resubmitting your figure files are available below the reviewer comments at the end of this letter. We look forward to receiving your revised manuscript. Kind regards, Christoph StrauchAcademic EditorPLOS Computational Biology Marieke van VugtSection EditorPLOS Computational Biology Feilim Mac GabhannEditor-in-ChiefPLOS Computational Biology Jason PapinEditor-in-ChiefPLOS Computational Biology  **Journal Requirements:** **Additional Editor Comments (if provided):** The reviewers (and we) have found the manuscript to be of interest. Yet, the expert reviews identify a list of issues that would need being addressed before reaching a final decision for the manuscript. Besides the comments given below, one reviewer further noted that they could not find the scripts implementing the simulations, please double-check/clarify in a revision.**Reviewers' comments:** Reviewer's Responses to Questions

**Comments to the Authors:**

Reviewer #1: This paper presents a significant advance over the standard modelling tools available to infer metacognitive sensitivity from confidence ratings. The authors build on their recent Psych Review paper, which devised a similar model for 2IFC confidence judgments, to show how fitting this model to rating data (the type 2 ROC) can recover two distinct parameters, confidence noise and confidence boost, that map onto distinct processes that have been posited in the confidence literature (eg metacognitive noise vs. additional information from post-decisional processing and other sources).

This will be a valuable tool and model for psychologists and neuroscientists interested in studying the underpinnings of confidence formation and metacognition. Accompanying the paper is a toolbox that should help this method gain widespread usage. The extension to model continuous ratings, and nonlinear mappings between subjective confidence and objective probabilities, is also impressive.

My main comment is about the extent to which the model and parameters can be effectively recovered – as establishing this is key to its advance beyond a single-parameter model like meta-d, and central to its practical usefulness. This comment can be split into a few different queries:

1) More general comment: in the figures, the focus is on parameter recovery, but not model recovery. The data are simulated from a two parameter (boost and noise) model, and these two parameters are recovered reasonably well (see next point). But this does not tell us whether there is the potential for confusion between model architectures – or whether the more complex model is justified in the first place. In other words, if you simulate data from a one-parameter model (only confidence noise, no boost – similar to classical SDT) then can this model be recovered (in comparison to the more complex two-parameter case)? And perhaps more importantly, if you simulate data from the new two-parameter model, can you show that this is preferred to a one-parameter noise-only model (over a reasonable range of parameter values)?

2) What counts as “good enough” parameter recovery here? It would be good to have some more quantitative metric of this. Eyeballing the plots, it looks like noise and efficiency are cleanly recovered in simulation, but that confidence boost is harder to pin down (eg Figs 4C, 6C), and also trades off against the noise parameter (which is another reason why quantifying confusability between single- and two-parameter architectures seems important). In Fig 7I, the error variance in boost recovery only comes down to acceptable levels in > 10,000 trials – which casts doubt on whether we should trust estimates obtained in fits to real datasets (eg as reported in the Appendix).

3) Following on from the previous point – in Figure 12D, inferred confidence boost is binary, being either 1 or 0 on different (sequential!) trials of the task. This seems implausible and also suggests the limited reliability of this parameter in real datasets.

4) On p. 9, it’s asserted that having more points on the type 2 ROC helps disentangle the confidence noise and confidence boost parameters. But why? Doesn’t figure 2 show that there are theoretical type 2 ROCs that are equivalent in boost and noise? How does having more points along these curves help with this?

Minor points:

- P. 3, bottom, I wasn’t convinced that you need to know the generative process to have a “proper measure of efficiency”. By analogy – d’ is a proper measure of sensitivity, but is compatible with a number of different process models of perception or memory etc.

- Eq. 3, is this a new draw of epsilon_s? Or is it the same sample used in Eq. 1? (I’m presuming the latter, but useful to make this explicit)

- Eq. 4, for a moment I thought that Dw was a new variable until I realised it was the product of D and w. Perhaps make this clearer using product notation.

- P. 7, what does “secondary information” mean here? (line 180)

- Line 488, “confidential” – I’m not sure this is the right word here. Perhaps “less influential”?

- I felt that the introduction omits consideration of recent attempts to innovate along the same lines as the CNBC model. Most notably, Boundy-Singer et al. Nat Hum Behav develop a model of metacognitive efficiency that can be fit across different stimulus strengths, and offers a process level interpretation of deficits in metacognition as failures to appropriately track sensory noise (meta-uncertainty). This model is cited in the discussion, but it was not clear how the authors see their efforts in relation to this. For instance, when reading the sentence “…but with no clear answer to the question of how to formally define and measure the efficiency with which human observers produce confidence judgments”, I found myself thinking “but what about Boundy-Singer and other attempts to do just this?”. Tackling this head on in the introduction would seem useful to ensure that the contribution of the paper is properly situated. How does the CNBC model go beyond this, or how does it take a different path?

- Again on framing – I really liked the extension of the model to incorporate nonlinearities in probability functions. It’s very nice that these parameters can be recovered independently of sensitivity, as it starts to provide a model-based formulation of metacognitive biases / overconfidence. It also integrates more closely the JDM / probability judgment and metacognition literatures. Perhaps this ambition can be flagged more clearly to the reader in the introduction.

- I wasn’t convinced that the ability to accommodate more than one stimulus strength is a major advance over meta-d (p. 13). It would be straightforward to fit a constant ratio to multiple meta-d/d points at different d’ values – and this was indeed done in some of the earlier work by Maniscalco & Lau.

Signed: Steve Fleming

Reviewer #2: The authors present a generative model of confidence judgments, the confidence noise and confidence boost (CNCB) model. The model is an alternative to approaches based on meta-d’ and assumes that, in addition to the sensory signal, confidence judgments can be further influenced by (1) information beyond the sensory signal can influence confidence, and (2) the confidence signal can be perturbed by its own source of noise. These two additional points of influence are represented as distinct model parameters, confidence boost (alpha) and confidence noise (sigma_c), respectively. Part of the motivation for developing the CNCB model is that it has a wider range of applications compared to meta-d’ in that CNCB can be used with multiple difficulty levels as opposed to just one.

An issue explored with the new model concerns tradeoffs between the confidence boost and confidence noise parameters. The authors show that the two parameters can be estimated well when there are more than two levels of confidence available for report and/or levels of stimulus strength (or task difficulty).

Parameter recovery analyses showed that the confidence boost and confidence noise parameters could be recovered well across a wide range of sample sizes and that the model could capture confidence efficiency across a range of circumstances. Further exploration shows a one-to-one correspondence between confidence efficiency as measured by the CNCB model and the m-ratio derived from the meta-d’ framework.

My evaluation of this manuscript is quite positive. The focus here is on the properties of the CNCB model and the authors have showcased how it is a potentially useful alternative to meta-d’. My main concerns are about the confidence boost parameter (alpha) and the relatively limited exploration of the CNCB’s parameter space. I discuss these two points below and include a third more minor point of confusion in the exposition.

1 – The effect of the confidence boost parameter is to reduce (or, when alpha = 1, entirely bypass) the influence of sensory noise on confidence (Equation 3). I think this is an interesting idea, but I am struggling to wrap my head around how to interpret this mechanistically (and if it is even appropriate to do so—it might not be and that’s fine). Is the parameter intended as something of a catch-all that captures all of the outside influences on the sensory signal for the purposes of making confidence judgments without committing to identifying exactly what they are? My sense is that the CNCB’s main use is as a vehicle for measuring confidence and confidence efficiency and so perhaps there is no need to think mechanistically about this parameter.

That said, even if confidence boost is intended to summarize “outside” effects on confidence, it would be helpful if the authors could describe experimental manipulations that might be expected to influence this parameter (the same would apply to confidence noise). For example, are there task conditions that would be expected to systematically influence confidence boost independent of confidence noise and vice-versa? The demonstrations of separable parameter estimation are important but a big question to me is whether the parameters can be independently manipulated experimentally.

2 – I think the demonstrations that multiple levels of confidence and stimulus strength (pp. 8-13) can help disentangle confidence boost from confidence noise are extremely interesting, but I was unsure what they added beyond the parameter recovery results reported on pp. 13-14. If I understand what was done correctly, aren’t these essentially small-scale parameter recovery studies? My sense is that a better approach might be to investigate how well parameters recover when generating values of confidence boost and noise are both randomly sampled as a function of the number of confidence levels/stimulus strengths. I think that doing things this way will provide a clearer picture of what is sketched in the current sections but it would also allow clearer examination of correlations across these parameters for fits to data.

3 – Page 7 Lines 173-182 – I had trouble following the argument in this passage. In describing the equivalence of the blue dot (simulated data) in Figure 2A with the green dot (ideal confidence observer) in Figure 2B, is this a general statement that the ideal confidence observer can produce equivalent performance to (any?) human/simulated observer, or is the argument about this one specific case?

In the end, I understood this passage to be illustrating a potential problem with estimating confidence boost and confidence noise separately (hence the demonstrations on pp. 8-13), but this was not clear to me upon first reading.

Line 117 – Typo, “…what the observer report[s] as their confidence…”

Reviewer #3: In this paper Mamassian & de Gardelle present their confidence noise confidence boost (CNCB) model for analysing confidence data. I think elements of the model are interesting, but there are aspects of the model and it’s behaviour I’m left unclear on.

Major comments:

1. There is something I don't understand in the model, specifically, in equation 3 (the specification of confidence evidence w). In the way that w is formulated, and as far as I can see, if a = 1 then the model says confidence has direct access to the stimulus evidence. This assumption cannot possibly hold. I would understand if confidence boost weighted the sensory evidence, i.e. w = (1-a)*(mu_s + e_s)…, so that when confidence boost is 1 the participant is estimating perceptual confidence blind to the sensory evidence. However, this doesn't seem the case. I appreciate that with the internal noise parameter w could be reformulated to get around the problem, but it wouldn’t change the conceptual issue.

2. A lot of the CNCB model is very similar to meta-d, to the extent that I’m struggling to see why and how they are not mathematically equivalent (or, indeed, why CNCB is theory driven while m-ratio is not). The paper (including in the introduction and discussion) would benefit from more discussion and exploration of how they differ (and when they’re the same), their respective costs and benefits, etc.

3. There were also some simple demonstrations of the model's behaviour that were missing. For example, what is predicted (in terms of efficiency, mostly) when type 1 sensitivity and bias are manipulated? Do you get the same efficiencies predicted when confidence bias is present (i.e. when the condition that confidence bins are used equally often is relaxed)? Some of this information was presented indirectly, late in the Results when comparing to m-ratio or applying the measures to continuous confidence data, but I think an explicit discussion of how the measures change with typical manipulations seen in experiments would be helpful. Similarly, it would be helpful to see how confidence noise, boost, and CNCB efficiency vary with factors like type 1 threshold and sensitivity on real data, and to compare CNCB efficiency against m-ratio with real data as well.

4. I struggled to map some of the results to the conclusions. In particular, on Line 250 (when showing how confidence noise and boost can be recovered with any number of confidence levels) it’s written:

"Unexpectedly, the estimation of these parameters is also possible even when there are only two confidence levels".

Perhaps I’m misreading the figures, but this doesn't seem to be the case for the blue parameters, which at 2 confidence levels are estimated far from the true value (noise of ~1.2 instead of 0.5 and boost of 1 instead of 0.2). Also, the IQRs around the yellow parameters in Fig 2C suggest to me that boost can't be meaningfully estimated for these parameters at all. Likewise, in Fig. 6 (recovering parameters over varying number of difficulty conditions) it seems to me that confidence noise is poorly recovered in most cases for the blue parameters (boost = 0.2, noise = 0.5), and that boost is poorly recovered for all 3 sets of parameters (either because the median is far off the true value, or because the IQR is very large, or both). Assuming I’ve understood correctly this seems to be a big problem, however if I’ve misunderstood could the manuscript be updated to clarify what these figures are showing?

Minor comments:

i. Thanks for sharing your toolbox. There's a typo in the doi - from what I could find the toolbox is available at 10.5281/zenodo.13348146, not 10.5281/zenodo.13348147

ii. A conceptual description of confidence boost and confidence noise would be really helpful, either in the introduction or results. What kind of psychological processes or manipulations would lead to one or the other increasing/decreasing?

iii. Eq3: is a unbounded? Fig 2B suggests it's bounded by 0 and 1 but from what I could see it wasn't specifically stated

iv. Fig 2. Legends are missing. Also Fig 4, 5A, C & D, Figs 6-9, Fig 11

v. I don't follow Figure 2B. If confidence boost & noise are varying on the x and y axis and all lines are representing participants with matched type 2 HR and FAR, what is being varied here to generate these different coloured lines? Or does each coloured line represent a participant with a different type 2 HR and FAR? If all lines correspond to an observer with a different type 2 HR and FAR, what determines the ideal observer?

vi. L188-189 Could you clarify what's meant by an 'extreme pair' here?

vii. Typo - L194 says "Below, we compare this CNCB efficiency measure to the M-ratio", but next section doesn't do this. Perhaps sections were ordered differently in an earlier draft?

viii. Doesn't the point made on L250 (about being able to recover parameters with only 2 levels of confidence) contradict what's said on L297-299 ("…We restricted the number of confidence levels to 2 (‘high’ and ‘low’) to verify that with only two stimulus strengths, confidence noise and confidence boost cannot be estimated (see Figure 2)."?

ix. L270-279 What are the confidence boost & noise parameters estimated from this simulation, and how do they differ when each level of confidence is estimated separately?

x. L401 "It is known that M-ratios are not fully independent of sensory criteria ([12]; Figure 9B). This is due in part to the fact that when there is a bias in sensory criterion, each sensory strength generates its own Type 2 ROC curve, and the meta-d’ computation needs to find a way to integrate these two incompatible sets of confidence into a single model." I have several about this:

- Barrett et al claim in that paper that m-ratio on degraded & enhanced signal models (ie when m-ratio ~=1) is effectively invariant to c. So it's true that that reference supports the claim that m-ratio isn't fully independent, but it's not quite true to the spirit of the claim. And indeed, the deviations with c are small (though they're substantial when you analyse response-conditional meta-ds). However when meta-d = d, m-ratio should be invariant to type 1 c by design.

- Following from that point, I'm struggling to understand why the blue line in Fig 9B isn't flat. Would the authors be able to speculate? Is the expected value of meta-d here 1?

- The IQRs for Fig 9B are missing

- Could you clarify what's meant by each sensory strength generating its own type 2 ROC? Do you mean that meta-d has to be estimated for each response class separately?

- I think that in its current form this sentence is somewhat misleading, because while it's true that the meta-d model is over-specified (causing this issue with calculating response-conditional meta-ds), CNCB is over-specified too (as shown in Fig 2). Could the authors explain why, despite both models being over-specified, CNCB doesn't end up with a bias but m-ratio does?

xi. To ensure that the results aren’t partly an outcome of how the data were simulated it would be helpful to see how CNCB efficiency and m-ratio covary, and vary with type 1 bias/sensitivity on real data.

xii. Assuming I haven’t missed it, I couldn't see a section on gamma recovery in Methods

xiii. L590-597. I'm not sure what the purpose of this self-consistency section is. It is an interesting idea, but isn't used in the paper as far as I can see.

xiv. I would have liked to see more advice for prospective users of the model in the discussion – what kind of data does CNCB efficiency work on, to what extent can a user evaluate effects of some manipulation on boost and noise separately, and are there any paradigm design choices the researcher should make to maximise their chance of getting clean results?

**Have the authors made all data and (if applicable) computational code underlying the findings in their manuscript fully available?**

Reviewer #1: Yes

Reviewer #2: Yes

Reviewer #3: **No: ** Ideally I would want to select 'not sure'. The toolbox code has been provided, however I cannot find the scripts that implement the simulations in this paper

PLOS authors have the option to publish the peer review history of their article (what does this mean? ). If published, this will include your full peer review and any attached files.

**Do you want your identity to be public for this peer review?** For information about this choice, including consent withdrawal, please see our Privacy Policy .

Reviewer #1: **Yes: ** Stephen Fleming

Reviewer #2: No

Reviewer #3: No

 **Figure resubmission:**While revising your submission, please upload your figure files to the Preflight Analysis and Conversion Engine (PACE) digital diagnostic tool, https://pacev2.apexcovantage.com/. PACE helps ensure that figures meet PLOS requirements. To use PACE, you must first register as a user. Registration is free. Then, login and navigate to the UPLOAD tab, where you will find detailed instructions on how to use the tool. If you encounter any issues or have any questions when using PACE, please email PLOS at figures@plos.org. Please note that Supporting Information files do not need this step. If there are other versions of figure files still present in your submission file inventory at resubmission, please replace them with the PACE-processed versions. 
---

## [Decision Letter · Decision Letter 1]

11 Mar 2025

PCOMPBIOL-D-24-01485R1

The Confidence-Noise Confidence-Boost (CNCB) model of confidence rating data

PLOS Computational Biology

Dear Dr. Mamassian,

Thank you for submitting your manuscript to PLOS Computational Biology. After careful consideration, we feel that it has merit but does not fully meet PLOS Computational Biology's publication criteria as it currently stands. Therefore, we invite you to submit a revised version of the manuscript that addresses the points raised during the review process.

Please submit your revised manuscript within 30 days May 11 2025 11:59PM. If you will need more time than this to complete your revisions, please reply to this message or contact the journal office at ploscompbiol@plos.org. Please include the following items when submitting your revised manuscript:

We look forward to receiving your revised manuscript.

Kind regards,

Christoph Strauch

Academic Editor

PLOS Computational Biology

Marieke van Vugt

Section Editor

PLOS Computational Biology

**Additional Editor Comments :**

R3 raises a couple of interesting points and requests for clarification that still need to be addressed (R1 and R2's points have all been comprehensively addressed). If the authors can clearly resolve these, the manuscript should be suitable for a minor revision without requiring further review.

**Reviewers' comments:**

Reviewer's Responses to Questions

Reviewer #1: Apologies for the delay in submitting my review. The authors have comprehensively tackled my previous concerns and I have no further comments.

Reviewer #2: The authors have addressed all of my major concerns. I am happy to recommend the manuscript be accepted for publication.

Reviewer #3: Thanks to the authors for their very thorough revisions. The model, results, and the usefulness of this approach are all much clearer to me now. I’ve only got a few more questions:

1. Thanks for adding that explanation of boost, it’s all clear to me now. You wrote that a possible manipulation that might target boost is varying the delay between Type 1 and 2 responses, but would a simpler one be presenting participants with (valid) information about the stimulus identity, or have I misunderstood? From what I gather, alpha is simply “non-sensory information about the stimulus”, so in the most extreme case where the experimenter tells the participant what presented stimulus was then boost should be 1.

2. Fig 7: These new figures are really informative. Even in the 1 parameter model (but also in the 2 parameter model), when confidence noise is less than around 0.5, parameter recovery is not great. Boost in particular can deviate a lot from the true parameter value too.

I suppose there will be few cases where a researcher would care about the raw values of those recovered parameters. What I expect will matter in most cases is any condition-wise differences in confidence noise/boost can be recovered.

I think a figure illustrating this would be very useful. If you vary how much d’, type 1 threshold or confidence noise changes between 2 conditions and keep all other parameters constant, does the difference in the recovered parameters correlate with the difference in the true parameters? I think the answer should be ‘yes’ but it’d be nice to see this.

3. On the topic of parameter recovery, in the results I don’t think there’s sufficient transparency about the difficulty in recovering boost. There are often statements such as “The simulations shown in Figure 11 indicate that the recovery of confidence noise and boost is quite robust over a large 500 range of confidence biases” (L 498-500), but the graphs aren’t reflecting that. For example, boost for the red parameters in Fig 11D is systematically underestimated and is poorly recovered for the yellow parameters. I think it’s important to be clear about the limitations of the model. In any case, despite boost being hard to recover CNCB efficiency behaves really well and confidence noise, mostly well.

4. On my initial reading of the paper I didn’t sufficiently appreciate the final section on continuous confidence ratings. Unless I am missing something, parameter recovery really benefits from using continuous confidence ratings - it seems that it’s with continuous ratings that the parameter recovery gets really good. Perhaps more importantly, that a CNCB user doesn’t need to bin their continuous confidence ratings into some arbitrary number of categories is a massive advantage of this model over meta-d. I think it’d be worth emphasising this more in the manuscript because this is a really valuable addition to the metacognition toolkit.

Minor comments

i. I agree with R1 that there should be a bit more discussion, either in the introduction or discussion, about alternatives to meta-d. Can your new sentence “We call this additional information the confidence boost, and note that this component is specific to our approach compared to other work [14].” Be elaborated on a bit more

ii. In my previous review I wrote (comment viii): “Doesn't the point made on L250 (about being able to recover parameters with only 2 levels of confidence) contradict what's said on L297-299 ("…We restricted the number of confidence levels to 2 (‘high’ and ‘low’) to verify that with only two stimulus strengths, confidence noise and confidence boost cannot be estimated (see Figure 2)."?

In your reply you wrote: “As discussed in point 4 above, the confidence noise and boost parameters should not be recoverable in theory. We believe that the fact that they are not completely random in practice comes from small response biases in the simulations.”

I’m afraid I still don’t understand this. Maybe my question was unclear – I was asking why the simulations in Fig 6 were run with only 2 levels of confidence (vs e.g 3 levels) when, at least in theory, it should not be possible to recover them.

iii. Typo on Line 464 “…since the original wok of Maniscalco and Lau, there *has [have] been other proposals…”

**Have the authors made all data and (if applicable) computational code underlying the findings in their manuscript fully available?**

Reviewer #1: Yes

Reviewer #2: Yes

Reviewer #3: Yes

PLOS authors have the option to publish the peer review history of their article (what does this mean? ). If published, this will include your full peer review and any attached files.

**Do you want your identity to be public for this peer review?** For information about this choice, including consent withdrawal, please see our Privacy Policy .

Reviewer #1: **Yes: ** Steve Fleming

Reviewer #2: No

Reviewer #3: No

**Figure resubmission:**
---

## [Editor Report · Decision Letter 2]

19 Mar 2025

Dear Dr. Mamassian,

We are pleased to inform you that your manuscript 'The Confidence-Noise Confidence-Boost (CNCB) model of confidence rating data' has been provisionally accepted for publication in PLOS Computational Biology.

Best regards,

Christoph Strauch

Academic Editor

PLOS Computational Biology

Marieke van Vugt

Section Editor

PLOS Computational Biology

---

## [Editor Report · Acceptance letter]

PCOMPBIOL-D-24-01485R2

The Confidence-Noise Confidence-Boost (CNCB) model of confidence rating data

Dear Dr Mamassian,

I am pleased to inform you that your manuscript has been formally accepted for publication in PLOS Computational Biology. Your manuscript is now with our production department and you will be notified of the publication date in due course.

With kind regards,

Zsofia Freund
